

# Intensive photochemical oxidation in the marine atmosphere:

# Evidence from direct radical measurements

Guoxian Zhang[1,2], Renzhi Hu[1, *], Pinhua Xie[1,2,3, **], Changjin Hu[1], Xiaoyan Liu[4],

Liujun Zhong[1], Haotian Cai[1], Bo Zhu[5], Shiyong Xia[5],Xiaofeng Huang[5], Xin Li[6],

Wenqing Liu[1]

[1] Key Laboratory of Environment Optics and Technology, Anhui Institute of Optics and Fine

Mechanics, HFIPS, Chinese Academy of Sciences, Hefei, China

[2] University of Science and Technology of China, Hefei, China

[3] College of Resources and Environment, University of Chinese Academy of Science, Beijing,

China

[4] School of Pharmacy, Anhui Medical University, Hefei, China

[5] Key Laboratory for Urban Habitat Environmental Science and Technology, School of

Environment and Energy, Peking University Shenzhen Graduate School, Shenzhen, China

[6] State Key Joint Laboratory of Environmental Simulation and Pollution Control, College of

Environmental Sciences and Engineering, Peking University, Beijing, China

**\*Correspondence to:** Renzhi Hu, Key Laboratory of Environment Optics and

Technology, Anhui Institute of Optics and Fine Mechanics, HFIPS, Chinese Academy

of Sciences, Hefei, China

**\*\*Correspondence to:** Pinhua Xie, University of Science and Technology of China,

Hefei, China

**Email addresses:** rzhu@aiofm.ac.cn (Renzhi Hu); phxie@aiofm.ac.cn (Pinhua Xie)



**Abstract:** Comprehensive observations of hydroxyl (OH) and hydroperoxy (HO$_2$) radicals were conducted in October 2019 at a coastal continental site in the Pearl River Delta (YMK site, 22.55°N, 114.60°E). The average daily maximum OH and HO$_2$ concentrations were (4.7–9.5) × 10$^6$ cm$^{-3}$ and (4.2–8.1) × 10$^8$ cm$^{-3}$, respectively. The synchronized air mass transport from the northern cities and the South China Sea exerted a time-varying influence on atmospheric oxidation. Under a typical ocean-atmosphere (OCM), reasonable measurement model agreement was achieved for both OH and HO$_2$ using a 0-D chemical box model incorporating the regional atmospheric chemistry mechanism version 2-Leuven isoprene mechanism (RACM2-LIM1). Land mass (LAM) influence promoted more active photochemical processes, with daily averages of 7.1 × 10$^6$ cm$^{-3}$ and 5.2 × 10$^8$ cm$^{-3}$ for OH and HO$_2$, respectively. Intensive photochemistry occurred after precursor accumulation, allowing local net ozone production comparable with surrounding suburban environments (5.52 ppb/h during the LAM period). The rapid oxidation process was accompanied by a higher diurnal nitrous acid (HONO) concentration (> 400 ppt). After a sensitivity test, HONO-related chemistry elevated the ozone production rate by 33% and 39% during the LAM and OCM periods, respectively, while the nitric acid and sulfuric acid formation rates were 52% and 35% higher, respectively. The simulated daytime HONO and ozone concentrations were reduced to a low level (~70 ppt and ~35 ppb) without the HONO constraint. This work challenges the conventional recognition of the MBL in a complex atmosphere. For coastal cities, the particularity of the HONO chemistry in the MBL tends to influence the ozone-sensitive system and eventually magnifies the background ozone. Therefore, the promotion of oxidation by elevated precursor concentrations is worth considering when formulating emission reduction policies.

**Keywords:** FAGE-LIF; OH and HO$_2$ radicals; Atmospheric oxidation; Marine boundary layer; Precursors;



# 1 Introduction

The marine boundary layer (MBL) occupies 71% of the planetary boundary layer, is a massive active carbon sink on Earth, and plays an irreplaceable role in coping with global climate change (Stone et al., 2012; Woodward-Massey et al., 2022b; Liu et al., 2022a). As a typical background atmosphere on the Earth, the MBL is equivalent to a natural smog chamber with limited anthropogenic emissions and is characterized by low NOx (the sum of nitric oxide (NO) and nitrogen dioxide ($NO_2$)) and non-methane hydrocarbons (NMHCs) under a layer of clean air (Woodward-Massey et al., 2022b). The lifetime of OH radical, a key oxidant, is on the order of a few hundred milliseconds (Fuchs et al., 2012). Due to the scarcity of oxidation precursors, including nitrous acid (HONO), formaldehyde (HCHO), and NMHCs, the reaction between $O^1D$ and water vapor generally dominates the radical initiation pathway in the marine environment. For example, in a tropical boundary layer observation experiment (reactive halogens in the marine boundary layer, RHaMBLe), ozone photolysis was found to account for 70% of the OH radical source based on the master chemical mechanism (MCM) (Whalley et al., 2010). The vital role of ozone photolysis is contrasting with typical polluted and semi-polluted areas investigated in a series of field campaigns, in which the propagation routes were found to dominate the radical source (Yang et al., 2021; Tan et al., 2019a). Therefore, studying the radical chemistry in the MBL provides a valuable opportunity to test the current understanding of atmospheric oxidation mechanisms in a natural setting.

Since the earliest observations off the coast of northern Norfolk in the Weybourne Atmospheric Observatory Summer Experiment in June 1995 (WAOSE95), more observations and simulations of radical chemistry in the MBL environment have been conducted using ground-based, airborne, and shipborne instruments (Qi et al., 2007; Kanaya et al., 2002; Kanaya et al., 2001; Mallik et al., 2018; Woodward-Massey et al., 2022a; Carpenter et al., 2011; Grenfell et al., 1999; Brauers et al., 2001; Whalley et al., 2010). Most field measurements have yielded well-reproduced OH and $HO_2$ concentration profiles via chemical mechanisms, with differences of within ~20%.



However, the base model is not sufficient to describe the radical chemistry in some
exceptional cases, especially in regard to the $HO_2$ radical. Considering the practical
association between halogen (Cl, Br, and I) chemistry and heterogeneous chemistry in
marine new particle formation, particularly the involvement of heterogeneous iodine-
organic chemistry, exploring the synchronous influence of these mechanisms on HOx
(OH and $HO_2$) radical chemistry in the MBL region is a worthy endeavor (Xu et al.,
2022; Huang et al., 2022). The mixing of air masses of continental and marine origins
can lead to more variability in radical concentrations. During seasonal measurements
of both OH and $HO_2$ in the Atlantic Ocean, variance analysis indicated that around 70%
of the variance of OH and $HO_2$ was due to diurnal behavior (in the form of photolysis
frequency), while the remaining variance was attributed to long-term seasonal cycles
(in the form of the changes in $O_3$, CO and air mass contribution) (Vaughan et al.,

93 2012).

The Chinese economy has undergone rapid development in recent years, and the
co-occurrence of primary and secondary regional pollution has become a severe
problem (Lu et al., 2019; Liu et al., 2022c). The interactions between air pollutants
from upwind cities, shipping vessels, and other anthropogenic emissions lead to
precursor accumulation (Sun et al., 2020; Zeren et al., 2022). The background ozone
concentration in key regions of China has increased year by year, highlighting the
significant influence of anthropogenic activities on the atmospheric oxidation in
background regions in China (Wang et al., 2009; Chen et al., 2022). However, little
research has been dedicated to the radical chemistry and oxidation mechanism in
regions with both coastal and continental features. To fill this research gap, in this
study, a field campaign was conducted on photochemistry in the MBL at a coastal site
in the Pearl River Delta. The OH and $HO_2$ radicals associated with other related
species were measured in October 2019, and the radical-related oxidation process was
identified to determine the photochemical efficiency in the marine atmosphere.
**2 Materials and methods**



## 2.1 Site description

As shown in Fig. 1(a), this observation campaign lasted for 11 days from October 18 to October 28, 2019, in Yangmeikeng (YMK, 22.55°N, 114.60°E), a coastal site in Shenzhen, Pearl River Delta. As the core city of the Greater Bay Area, Shenzhen is bordered by Dongguan to the north, Huizhou to the east, and Hong Kong to the south. The YMK site is on the Dapeng Peninsula, to the southeast of Shenzhen, between Mirs Bay and Daya Bay. As it is adjacent to the port of Hong Kong, precursors from ship emissions may influence the atmospheric chemistry. The site is approximately 35 m above sea level, and the sea is approximately 150 m to the east. No apparent local emissions exist, and the surrounding forest is lush (Fig. 1(b)). In addition to anthropogenic and vegetation emissions, the site is also affected by the synchronization of plumes from northern cities and the South China Sea (Niu et al., 2022; Xia et al., 2021). Due to its significant time-varying pollution characteristics, this area is an ideal site for studying the effects of plume transport on atmospheric oxidation.

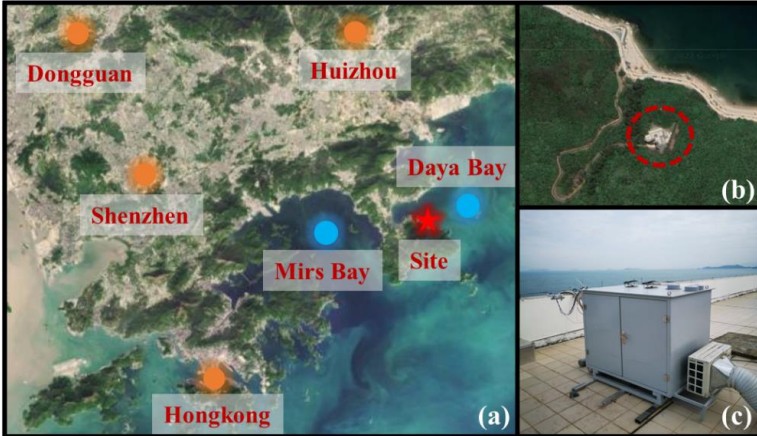

**Fig. 1.** Details of the observation site**(a)** The location of the measurement site and surrounding cities. The satellite map data is extracted from © Google Earth.**(b)** Th close shot of the measurement site location.**(c)** The actual image for the LIF-Box.

## 2.2 Instrumentation

### 2.2.1 HOx radical measurements

The OH and $HO_2$ radicals were measured via laser-induced fluorescence (LIF).



The OH radical can be directly measured by exciting the fluorescence using a 308-nm
laser. $HO_2$ is converted into the OH radical via chemical transformation and then
detected in the form of OH radical. The self-developed instrument, the Anhui Institute
of Optics Fine Mechanics-LIF (AIOFM-LIF), was used to conduct the measurements
(Zhang et al., 2022a; Wang et al., 2021; Wang et al., 2019). This system has been used
in key regions of China, including the Yangtze River Delta, Pearl River Delta, and
Chengdu-Chongqing region, and achieved good performance in a comparison
experiment with a LIF system jointly developed by Forschungszentrum Jülich and
Peking University (PKU-LIF) (Zhang et al., 2022b).
The system and detection interference process have been described in detail in
previous studies (Zhang et al., 2022a). Briefly, the system consists of a laser output
module, a radical detection module, and a control and data acquisition module. These
modules are integrated into a sampling box with constant temperature and humidity
control (Fig. 1(c)). The laser output module is a union of an Nd:yttrium-aluminum-
garnet (YAG) solid-state laser, a 532-nm laser output, and a tunable dye laser. In the
radical detection module, the OH and $HO_2$ fluorescence cells are combined in parallel
and share a common axial optical path. The 308-nm laser is introduced into the $HO_2$
cell first and then into the OH cell via an 8-m fiber. To maintain the detection
efficiency, the power in the OH fluorescence cell should be at least 15 mW. In the
detection process, a set of lenses was deployed and positioned in front of the
microchannel plate detector (MCP) to boost the fluorescence collection capacity. Each
MCP detector contains a timing control instrument to optimize the signal-to-noise
ratio (SNR) of the fluorescence detection. Efficient ambient air sampling was
achieved using an aluminum nozzle (0.4 mm orifice), and the pressure in the chamber
was maintained at 400 Pa via a vortex vacuum pump (XDS35i, Edwards) to avoid
fluorescence quenching.
A wavelength modulation for the background measurement that periodically
switches from an on-resonant state to a non-resonant state has been widely used to
obtain spectral zero. The ozone photolysis interference was subtracted according to





laboratory experiments. An OH measurement comparison with an interference-free instrument, PKU-LIF, was conducted in a real atmosphere in a previous study (Zhang et al., 2022b). The ozonolysis interference on the measurement consistency of both systems was excluded under high-NOx and high-NMHC conditions, confirming the general applicability under complex atmospheric pollution. For $HO_2$ measurement, the NO concentration corresponding to a conversion efficiency of ~15% was selected to avoid $RO_2 \rightarrow HO_2$ interference (especially from $RO_2$ radicals derived from long-chain alkanes (C ≥ 3), alkenes, and aromatic hydrocarbons).

A standard HOx radical source based on the simultaneous photolysis of $H_2O/O_2$ by a 185 nm mercury lamp was used to complete the calibration of the detection sensitivity (Wang et al., 2020). During the observation campaign, the instrument was calibrated every 1 or 2 days (except for shutdown during rainy periods), and the sensitivity used for the data processing was an average of all of the calibration results. Considering the system error and calibration error, the detection limits of the OH and $HO_2$ radicals were $3.3 \times 10^5$ cm$^{-3}$ and $1.1 \times 10^6$ cm$^{-3}$ (60 s, 1σ), respectively, at a typical laser power of 15 mW, and the measurement errors were 13% and 17%, respectively.

**2.2.2 Supporting measurements**

In addition to measuring the HOx radicals, an extensive suite of relevant species was also measured close to the LIF instrument to improve the analysis of the radical photochemistry. Detailed information about the measurement instrument is presented in Table S1, including the meteorological parameters (wind speed (WS), wind direction (WD), temperature (T), relative humidity (RH), pressure (P), and solar radiation (J-values)), conventional pollutants (ozone ($O_3$), carbonic oxide (CO), and sulfur dioxide ($SO_2$)), secondary pollution precursors (HONO, NO, $NO_2$, HCHO, and NMHCs), and destruction products (particulate matter ($PM_{2.5}$)). In addition to HCHO, other volatile organic compounds (VOCs) were detected using a gas chromatograph coupled with a flame ionization detector and mass spectrometer (GC-FID-MS). Ninety-nine types of VOCs, including C2–C11 alkanes, C2–C6 alkenes, C6–C10



aromatics, halohydrocarbons, and some oxygenated VOCs (OVOCs), were observed
using the GC-FID-MS at a 1-h time interval. Only isoprene was considered as a
representative of biogenic VOCs (BVOCs). All of the instruments were located close
to the roof of the fourth floor, nearly 12 m above the ground to ensure that all of the
pollutants were located in a homogeneous air mass.

**2.3 Model description**

A 0-D chemical box model incorporating a condensed mechanism, the regional
atmospheric chemistry mechanism version 2-Leuven isoprene mechanism (RACM2-
LIM1), was used to simulate the radical concentrations and the generation of radical-
related secondary pollution (Stockwell et al., 1997; Griffith et al., 2013; Tan et al.,
2017). The meteorological parameters, conventional pollutants, and precursor
concentrations mentioned in Section 2.2.2 were input into the model as boundary
conditions. All of the constraints were unified to a temporal resolution of 15 min
through averaging or linear interpolation. The overall average during the observations
was substituted for large areas of missing data due to instrument maintenance or
failure. Three days of data were entered in advance as the spin-up period, and a
synchronized time-dependent dataset was eventually generated. The hydrogen ($H_2$)
and methane ($CH_4$) concentrations were set to fixed values of 550 ppb and 1900 ppb,
respectively. The physical losses of species due to processes such as deposition,
convection, and advection were approximately replaced by an 18 h atmospheric
lifetime. According to the measurement accuracy, the simulation accuracy of the
model for the OH and $HO_2$ radicals was 50% (Zhang et al., 2022a).
Considering the environmental characteristics of the MBL, the gas-phase
mechanisms for bromine ($Br_2$) were introduced into the base model to diagnose the
impacts of the reactive bromine chemistry at the field site. The details of the
mechanisms involved are listed in Tables S2 and S3. The halogen species were not
available in the YMK site, so the $Br_2$ concentration during the same season at a
coastal site in the Pearl River Delta was used as a reference value (average daytime
concentration of 3–5 ppt at a coastal ground site in Hong Kong, China).



# 3 Results

## 3.1 Meteorological and chemical parameters

### 3.1.1 Data overview

Fig. S1 presents the time series of the main meteorological parameters and pollutants during the observation period at the YMK site. Except for on 2 days, October 26 and 28, the meteorological characteristics of the other days were generally stable. The daily maximum T, RH, and J-values did not vary significantly. The suitable temperature (20–30°C) and humidity (50–80%) conditions promoted the stable oxidation of the diurnal photochemistry. The peak $j(O^1D)$ value was approximately $2.0 \times 10^{-5}$ $s^{-1}$, exhibiting the typical characteristics of intense light radiation in autumn in the Pearl River Delta region (Yang et al., 2022a; Tan et al., 2022).

As typical marine air components, the concentrations of NOx, CO, $PM_{2.5}$, and other pollutants were lower than those detected in other observation campaigns in both urban and suburban areas in the Pearl River Delta region (Tan et al., 2019b; Lu et al., 2012; Yang et al., 2022b). The $PM_{2.5}$ and CO concentrations exhibited good consistency and even mild pollution features on some dates, reflecting the influence of human activities. Contrary to the conventional belief that marine ozone is a global background setting, the ozone concentration in the YMK site was always at the critical value of the updated Class I standard (GB3095-2012, average hourly $O_3$ of 81 ppb at 25°C and 1013 kPa). The occurrence of fewer emissions reduced the titration effect, resulting in the ozone exhibiting no apparent diurnal trend on some of the dates and a high background value at night (67.3 ± 7.6 ppb). The NOx concentrations also maintained typically low levels on most dates. The daily maximum NMHC concentration peaked at 19.3 ± 3.0 ppb, and the maximum value of ~40 ppb occurred on October 27. Local biological emissions significantly affected the NMHC composition of the site, and isoprene, a representative BVOC, achieved a noon maximum of 0.82 ± 0.16 ppb. Neither anthropogenic alkenes nor aromatic




hydrocarbons were abundant, and OVOCs accounted for approximately 50% of the
total. As a photochemical indicator, formaldehyde peaked at ~4 to ~8 ppb on October
18, 19, and 27, suggesting a more vigorous oxidation process. HONO exhibited a
grooved distribution with high daytime (0.49 ± 0.097 ppb) and low nighttime (0.20 ±
0.11 ppb) concentrations. This unique distribution of HONO has been observed in
remote environments in several previous observation campaigns (Jiang et al., 2022;
Crilley et al., 2021). An extremely high daytime HONO concentration will
significantly affect the chemical composition of the atmosphere and the secondary
pollution generation.
**Table 1.** Summary of radical concentrations and related species concentrations at MBL. All data are listed as the average in noontime (10:00~15:00).

| Campaign | Location | Date | OH ($10^6$cm$^{-3}$) | HO$_2$ ($10^8$cm$^{-3}$) | HCHO (ppb) | HONO (ppb) | NOx (ppb) | O$_3$ (ppb) | Ref |
|---|---|---|---|---|---|---|---|---|---|
| WAOSE95 | Weybourne, UK | 1995 (Jun) | 5.0 | - | 1.50 | 0.10 | <2.0 | 40.0 | (Grenfell et al., 1999) |
| ALBATROSS | Atlantic Ocean | 1996 (Oct-Nov) | 7.0 | - | 0.50 | - | - | 25.0 | (Brauers et al., 2001) |
| EASE96 | Mace Head, Ireland | 1996 (Jul-Aug) | 2.3 | 2.6 | - | - | ~1.0 | 45.0 | (Carslaw et al., 1999) |
| EASE97 | Mace Head, Ireland | 1997 (Apr-May) | 1.8 | 1.0 | 0.70 | - | 0.95 | 46.0 | (Creasey et al., 2002) |
| ORION99 | Okinawa Island, Japan | 1999 (Aug) | 4.0 | 4.3 | - | 0.20 | 6.3 | 23.0 | (Kanaya et al., 2001) |
| RISOTTO | Rishiri Island, Japan | 2000 (June) | 7.4 | 3.1 | - | - | 0.45 | - | (Kanaya et al., 2002) |
| RISFEX | Rishiri Island, Japan | 2003 (Aug) | 2.7 | 1.5 | - | - | 0.2 | 28.0 | (Qi et al., 2007) |
| RHaMBLe | Cape Verde, Atlantic Ocean | 2007 (May-Jun) | 9.0 | 6.0 | 0.30 | - | 0.014 | 35.0 | (Whalley et al., 2010) |
| SOS | Cape Verde, Atlantic Ocean | 2009 (Jun; Sep) | 9.0 | 4.0 | 1.9 | - | 0.050 | 40.0 | (Carpenter et al., 2011) |
| CYPHEX | Cyprus, Mediterranean | 2014 (Jul) | 5.8 | 6.3 | ~1.0 | ~0.080 | <1.0 | 69.0 | (Mallik et al., 2018) |
| ICOZA (NW-SE) | North Norfolk, UK | 2015 (Jul) | 3.0 | 1.4 | 0.9 | 0.052 | 2.0 | 39.0 | (Woodward-Massey et al., 2022b) |
| ICOZA (SW) | North Norfolk, UK | 2015 (Jul) | 4.1 | 1.0 | 1.1 | 0.097 | 3.0 | 31.0 | (Woodward-Massey et al., 2022b) |
| HT | Hok Tsui, China | 2020 (Oct-Nov) | 4.9 | - | 1.0 | 0.15 | ~4.0 | 65.0 | (Zou et al., 2022) |
| YMK (Land Mass) | Shenzhen, China | 2019 (Oct) | 7.1 | 5.2 | 3.4 | 0.66 | 6.4 | 75.6 | This work |
| YMK (Ocean Mass) | Shenzhen, China | 2019 (Oct) | 4.5 | 4.9 | 1.2 | 0.48 | 3.0 | 78.1 | This work |


**3.1.2 Influences of different air masses**
During the YMK observation campaign, the wind direction was mainly easterly





and southerly, and the wind speed was below 3 m/s. The conventional wind direction
is insufficient to reflect the air mass trajectory at a slightly higher altitude due to the
mountain-valley breeze (Niu et al., 2022). Using the hybrid single-particle Lagrangian
integrated trajectory (HYSPLIT) model, the 24-h backward trajectories on special
days were obtained (Fig. S2). In Fig. S2, the red, blue, and green trajectories represent
the results at altitudes of 100, 500, and 1000 m above ground level, respectively. Two
typical transportation pathways dominated the air parcels. One originated from the
northern megacities in the Pearl River Delta (defined as the land mass, LAM),
especially on October 18, 19, and 27. In contrast, a clean air mass from the east or
northeast was mainly transported to the observation site from the ocean (defined as
the ocean mass, OCM), with representative episodes on October 22, 25, and 26.
Serval observation campaigns have discovered the relationship between wind
direction and radical chemistry (Lu et al., 2012; Fuchs et al., 2017; Niu et al., 2022).
Although there was no apparent wind speed condition, the dominant air mass still
influenced the pollutant concentrations due to the particularity of the marine site.
During the OCM period, the NOx and HCHO concentrations exhibited relatively
clean characteristics that were consistent with those previously reported (Table 1).
However, both the HONO and $O_3$ concentrations were twice as high as those of the
other components, and their daily average values (10:00–15:00) reached 0.48 ppb and
78.1 ppb, respectively. Compared with the OCM period, the meteorological
conditions (T, RH, and J-values) changed greatly during the LAM episode. The
pollutants were accumulated due to the transport of the plume from the northern cities
(Fig. 2). Both NO and $NO_2$ peaked at around 10:00, exhibiting prominent pollution
characteristics. The diurnal peaks of the HONO and HCHO concentrations were much
higher than those of the Integrated Chemistry of Ozone in the Atmosphere (ICOZA)
Project observations (a pollution period dominated by a southwest wind direction)
(Woodward-Massey et al., 2022b). The HONO concentration was 6.8 times higher
than when the wind direction was southwest in the ICOZA observations, while the
HCHO concentration was 3.1 times higher. The abundance of oxidation precursors





(HONO, HCHO, O₃, and NMHCs) reflected the unique atmospheric conditions in the
marine environment in China, which originated from the complex atmospheric
pollution.

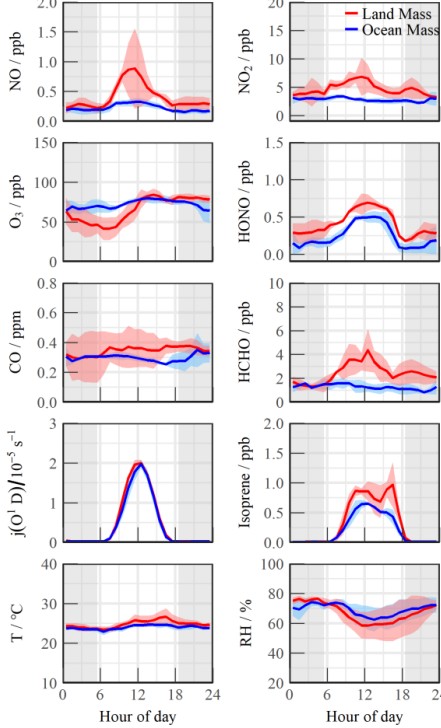

**Fig. 2.** Mean diurnal profiles of measured trace gases parameters during Land mass and Ocean mass episodes. The
coloured shadows denote the 25 and 75% percentiles. The grey areas denote nighttime.
**3.2 HOx radical concentrations and modeled OH reactivity**

296         Fig. 3(a) and (b) shows the time series of the simulated and observed OH and

HO₂ radical concentrations during the observation campaign. The time series of the
simulated OH reactivity ($k_{OH}$) is presented in Fig. 3(c). The observed OH and HO₂
radicals exhibited significant diurnal trends. The average daily maximum OH and
HO₂ values were $(4.7–9.5) \times 10^6$ cm$^{-3}$ and $(4.2–8.1) \times 10^8$ cm$^{-3}$, respectively. The
peak $k_{OH}$ value was commonly less than 10 s$^{-1}$. Due to human activities, the simulated
$k_{OH}$ reached more than ~15 s$^{-1}$ on some days. The radical concentrations and
reactivity exhibited similar trends, which differed from reports on urban and semi-
urban areas where inorganic species (NOx and CO) were the dominant controllers of



$k_{OH}$ (Zhang et al., 2022a; Tan et al., 2019b; Lou et al., 2010).

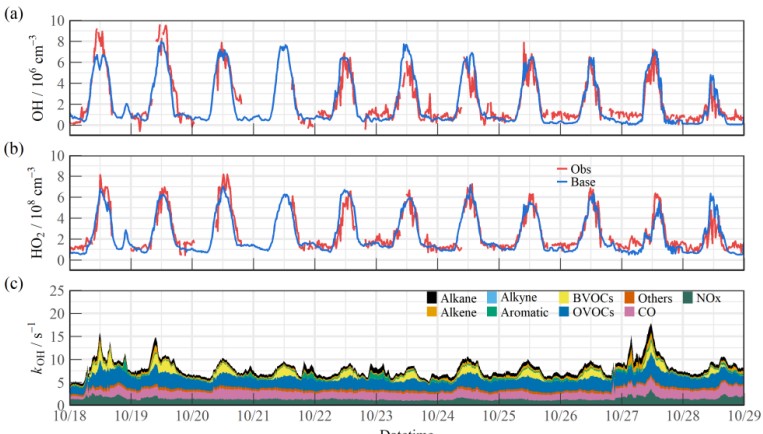

**Fig. 3.** Timeseries of the observed and modelled parameters for OH, HO₂ and $k_{OH}$ during the observation
308                        period. **(a)** OH, **(b)** HO₂, **(c)** $k_{OH}$.

The OH and HO₂ concentrations were calculated using a base model
incorporating the RACM2-LIM1 mechanism. Overall, the observed OH and HO₂
concentration data were both well reproduced by the base model (Fig. 4(a)–(b)). The
base model slightly overestimated the OH radical, suggesting that a radical removal
pathway was missing. Halogen species have been recognized as potent oxidizers that
can boost photochemistry (Xia et al., 2022; Peng et al., 2021). A sensitivity test was
performed by imposing ~3 ppt Br₂, a typical mixing ratio reported for a coastal site in
the Pearl River Delta, into the base model to diagnose the impact of the halogen
chemistry on the troposphere chemistry (Xia et al., 2022). The details of the
mechanisms involved are listed in Tables S2 and S3. In this scenario (Fig. 4(a)–(b),
green line), the simulated OH was 11.6% lower than in the base model, and no
significant effect on the HO₂ radical was identified. The daily maximum calculated
total OH reactivity was 9.9 s⁻¹ (Fig. 4(c)). Regarding the contributions of the
inorganic species, the contributions of CO and NOx were close at 18.0% and 14.8%,
respectively. Nearly 70% of the reactivity was accounted for by the organic species,
among which the OVOCs were the largest contributor (30.6%). The anthropogenic
alkanes, alkenes, and aromatic hydrocarbons contributed less than 10% to the
reactivity in the marine environment. The BVOCs emitted by the surrounding forest





could not be ignored, accounting for 15.7%.

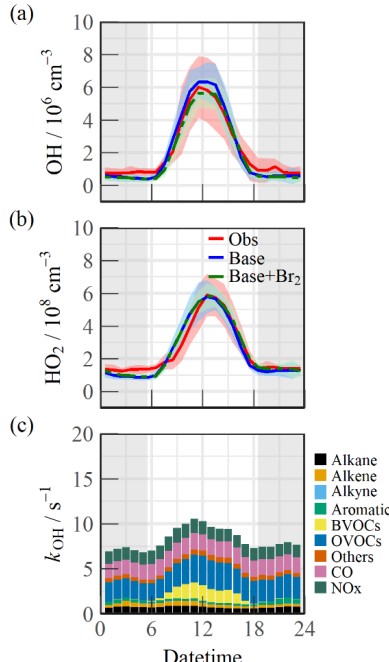

**Fig. 4.** Median diurnal profiles of the observed and modelled parameters for OH, HO$_2$ and $k_{OH}$ during the observation period. **(a)** OH, **(b)** HO$_2$, **(c)** $k_{OH}$.

The regional transport of radicals was generally impossible due to their short
lifetimes. However, the air mass transport of the precursors increase the ROx primary
sources. Under the linkage of NO concentration, this leads to accelerated cycling
efficiency of the radicals, promoting the accumulation of photochemical products.
The effects can be seen directly in the changes to the oxidation level. Isoprene is
discussed as an example. The prevailing wind direction experienced a series of
southerly-easterly shifts from 8:00 to 18:00 on October 18 (Fig. 5(a)). The growth and
decline of the isoprene concentration were highly correlated with the changes in the
wind direction (Fig. 5(b)), and the maximum concentration (2.1 ppb) occurred at
17:00 under the southwest wind. Correspondingly, the sensitive LIF instrument
captured the decrease in the concentration at noon. The base model simulated the
fluctuations in the OH concentration, but the solar radiation did not vary, indicating
that the change in the precursor accelerated the instantaneous OH → HO$_2$ propagation
(Fig. 5(c)). In addition, the evolution of the air mass composition inhibited the





conversion of $HO_2$ to OH and maintained the high $HO_2$ level during the afternoon
(Fig. 5(d)).

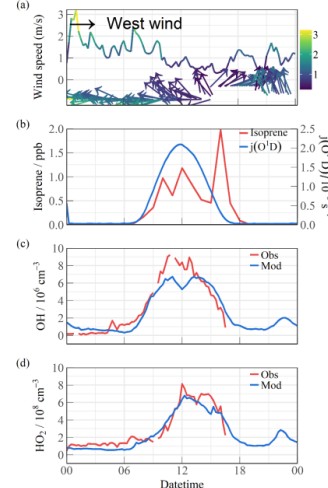

**Fig. 5.** Median diurnal profiles of the observed and modelled parameters during a typical case of rapid wind
direction change on October 18. **(a)** Wind direction and speed, **(b)** Isoprene concentration and solar radiation
$(j(O^1D))$, **(c)** The observed and modelled OH concentration, **(d)** The observed and modelled $HO_2$ concentration.
Therefore, it is worth comparing the concentrations and reactivities of the radicals
by classifying the predominant air mass (Fig. 6). During the OCM period, the
observed OH and $HO_2$ radicals could be reflected by the base chemical mechanism,
with daily averages of $4.5 \times 10^6$ cm$^{-3}$ and $4.9 \times 10^8$ cm$^{-3}$, respectively. Compared to
other campaigns (Table 1), the observed maximum values were within reasonable
ranges (OH: $2$–$9 \times 10^6$ cm$^{-3}$; $HO_2$: $1$–$6 \times 10^8$ cm$^{-3}$). Despite low NOx levels during
the OCM period, the $HO_2$ radical was not overestimated using the base model, which
was dissimilar to many MBL observations. The heterogeneous uptake pathway did not
need to be further investigated due to the low $PM_{2.5}$ concentration during the OCM
period ($< 25$ µg/m$^3$). However, both the OH and $HO_2$ radical concentrations reached
higher levels during the LAM-dominant period, indicating a more active
photochemical process. The diel averages for the OH and $HO_2$ radicals were $7.1 \times 10^6$
cm$^{-3}$ and $5.2 \times 10^8$ cm$^{-3}$, respectively, which were notably higher than the levels
reported in the ICOZA observations (Woodward-Massey et al., 2022b). The base
model underestimated both the OH and $HO_2$ concentrations between 10:00 and 15:00,
and the observation-to-model ratio was greater than 1.2. Compared with the OCM-



dominant episode, the higher reactivity during the LAM period indicated the
occurrence of efficient recycling during the ROx propagation (12.4 s$^{-1}$ vs. 8.8 s$^{-1}$).
The higher contributions of the BVOCs (only isoprene was considered, 15.6%) and
OVOCs (30.2%) to the reactivity reflected the diverse composition of the VOCs in the
forest environment. The more reactive atmosphere did not introduce a missing OH
source in the afternoon, but radical cycling under enhanced photochemistry is worth
discussing (Hofzumahaus et al., 2009). As a representative of the OVOCs, HCHO
reflects the photochemical level to a certain extent. As shown in Fig. S3, a solid
positive dependence between the OH$_{obs}$-to-OH$_{mod}$ ratio and HCHO was observed (the
daytime data were restricted according to j(O$^1$D) > 5 × 10$^{-6}$ s$^{-1}$). Considering the
essential contributions of the OVOCs and BVOCs during ROx recycling, the other
unmeasured species (mono-terpenes and reactive halogens) involved in the oxidation
cycle were responsible for the elevated photochemistry. Obtaining the full magnitude
of the radical-related parameters is necessary to compensate for the discrepancy in the
concentration closure experiments.

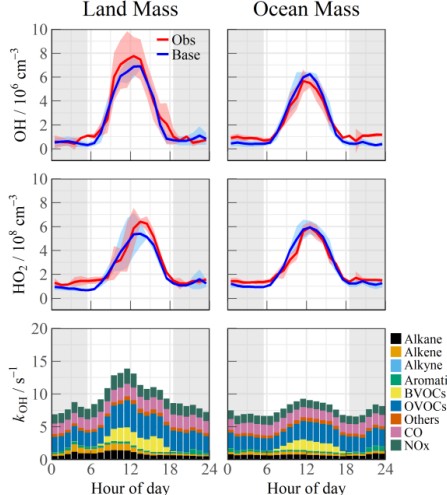

**Fig. 6.** Median diurnal profiles of the observed and modelled OH, HO$_2$, $k_{OH}$ during LAM and OCM episodes.
The coloured shadows for OH and HO$_2$ radicals denote the 25 and 75% percentiles. The grey areas denote
nighttime.
# 4 Discussion
## 4.1 Experimental radical budget balance





**4.1.1 OH radical**

A process-oriented experiment was conducted to investigate the photochemistry progress from a budget balance perspective (Woodward-Massey et al., 2022a; Tan et al., 2019b; Yang et al., 2021). The OH was in a photostationary steady state due to its short lifetime. The total OH removal rate was directly quantified from the union of the OH concentration and the reactivity (R (1)):

$$D(OH) = [OH] \times k_{OH}. \tag{1}$$

The total production rate of the OH radical was the sum of the primary sources (O$_3$/HONO photolysis and ozonolysis reactions) and secondary sources (HO$_2$ + NO) (R (2)):

$$P(OH) = j_{HONO}[HONO] + \varphi_{OH} j(O^1D)[O_3] + \Sigma i \left\{ \varphi_{OH}^i k_{Alkenes+O_3}^i [Alkenes][O_3] \right\}$$

$$+ (k_{HO_2+NO}[NO] + k_{HO_2+O_3}[O_3])[HO_2]. \tag{2}$$

The diel profiles of the experimental OH budget during the LAM and OCM periods are shown in Fig. 7. Both the observed OH and HO$_2$ radicals were introduced into the budget calculations. Because $k_{OH}$ was not measured during the observation experiment, the simulated value was used as the lower limit to analyze the removal rate (Yang et al., 2022b). During the OCM period, the HO$_2$ + NO reaction accounted for ~50% of the OH yield. The maximum of 6.6 ppb/h occurred at around 12:00. The photolysis reactions could increase the daytime contributions of HONO and O$_3$ to 1.52 ppb/h and 0.84 ppb/h, respectively (10:00–15:00). The contribution of the non-photolytic radical source (ozonolysis reactions) was almost negligible.



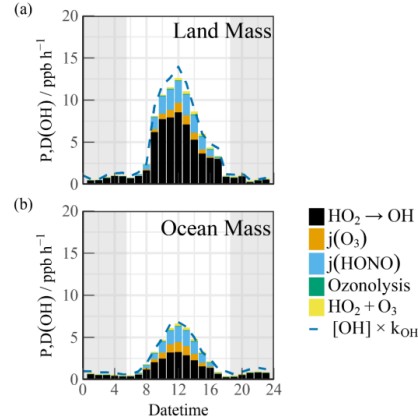

**Fig. 7.** The diurnal profiles of the experimental OH budget during **(a)** Land mass and **(b)** Ocean mass
episodes. The blue line denotes the OH destruction rate([OH]×$k_{OH}$). The grey areas denote nighttime.
Compared with other marine observations, the calculated OH generation rate was
approximately twice that reported in the ICOZA Project and five times that obtained
in the RHaMBLe Project (Woodward-Massey et al., 2022a; Whalley et al., 2010).
During the LAM period, the OH generation rate reached a maximum of 12.6 ppb/h,
accompanied by a secondary source contribution of 67% (from the reaction between
$HO_2$ and NO) during the daytime, which was close to several observations related to
polluted plumes (Woodward-Massey et al., 2022a; Tan et al., 2019b; Lu et al., 2012;
Yang et al., 2022b). No additional OH radical source was needed when the simulated
$k_{OH}$ was introduced into the experimental budgets. The difference between P(OH)
and D(OH) was less than 2 ppb/h, indicating the absence of a nontraditional OH
recycling pathway (X mechanism) under low NO concentration conditions
(Hofzumahaus et al., 2009).
**4.1.2 Total ROx radicals**
The budget analysis of the $HO_2$ and $RO_2$ radicals could not be performed well
due to the lack of $RO_2$ radical observation data. The diurnal profiles of the ROx
production (P(ROx)) and termination rate (L(ROx)) for the different air masses are
shown in Fig. 8. The P(ROx) could reach 3.36 ppb /h with an ocean plume. HONO
photolysis controlled nearly half of the primary sources (45.7%), and the daily
distribution was consistent with that of solar radiation. The ozone-related



contributions from photolysis and ozonolysis were approximately 25.1% + 11.5%.
The remaining contribution was from the photolysis of carbonyls (HCHO and
OVOCs) (15.0%). The anthropogenic contribution to the radical chemistry was not
ignorable, and the ROx source in this observation was exponentially higher than that
in other MBL observations (Woodward-Massey et al., 2022a; Stone et al., 2012;
Whalley et al., 2010; Mallik et al., 2018). The P(ROx) of the LAM was close to that
in Shenzhen (~4 ppb/h) but was significantly lower than that in Yufa (~7 ppb/h) and
the BackGarden (~11 ppb/h) (Tan et al., 2019b; Lu et al., 2012; Yang et al., 2022b).
The reactions between ROx and NOx and self-combination were the main pathways
of radical termination (~70%). The contribution of the formation of peroxynitrite to
the L(ROx) could not be ignored in the daytime.

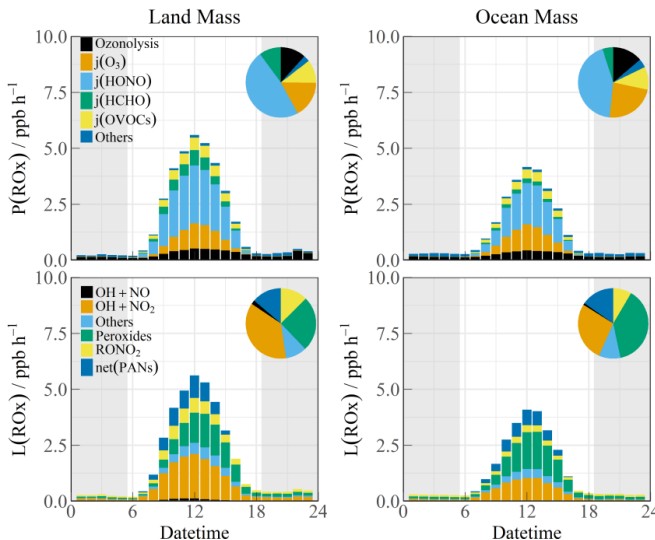

**Fig. 8.** The diurnal profiles of ROx budget during Land mass and Ocean mass episodes. The pie chart denotes
proportions in different parts during the daytime (10:00-15:00). The grey areas denote nighttime.
Due to the high HONO concentration during the daytime, the photolysis reaction
made daytime contributions of 1.52 ppb/h and 2.19 ppb/h during the OCM and LAM
periods, respectively. As the only known gas-phase source, OH + NO accounted for a
negligible proportion of the HONO loss. Considering the location of the YMK site,
HONO from cruise ship emissions is a possible component of the primary
anthropogenic source (Sun et al., 2020). Other active tropospheric HONO sources


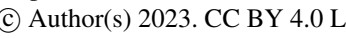


(heterogeneous reactions with $NO_2$ and $p(NO_3^-)$ photolysis) are worthy of
consideration and significantly contribute to the atmospheric oxidation in the MBL
area (Zhu et al., 2022; Crilley et al., 2021).

**4.2 Local ozone production rate**

Peroxyl radical chemistry is the essential photochemical source of tropospheric
ozone (F(Ox), R (3)):
$$F(O_x) = k_{HO_2+NO}[NO][HO_2] + \sum_i k_{RO_2^i+NO}[NO]RO_2^i. \qquad (3)$$

NO reacts with $HO_2$ and $RO_2$ radicals to form $NO_2$, and then, photolysis occurs to
form $O_3$ under solar radiation. $NO_2$ and ozone are the two sides of the oxidation
reservoir. The effect of local emissions on the photodynamic equilibrium can be
avoided by characterizing the photochemical production of the total oxidants (Tan et
al., 2019b). Ox is mainly photochemically removed through ozone photolysis,
ozonolysis, radical chain propagation (OH/$HO_2$ + $O_3$), and chain termination (OH +
$NO_2$) reactions in the troposphere (D(Ox), R (4)):
$$D(O_x) = \varphi_{OH}j(O^1D)[O_3] + \Sigma i \left\{k_{Alkenes+O_3}^i[Alkenes][O_3]\right\} + (k_{O_3+OH}[OH] +$$

$$k_{O_3+HO_2}[HO_2])[O_3] + k_{OH+NO_2}[OH][NO_2]. \qquad (4)$$

The net formation rate (P(Ox)) can be calculated by subtracting D(Ox) from F(Ox):
$$P(O_x) = F(O_x) - D(O_x). \qquad (5)$$

The simulated $RO_2$ radical concentration was introduced into the F(Ox)
calculation. The diurnal variations in the ozone generation in the different air masses
are shown in Fig. 9. The contribution of the $HO_2$ radical to F(Ox) was approximately
60%. The $RO_2$ radicals consisted of various types such as methyl peroxyl ($MO_2$),
acetyl peroxy radicals ($ACO_3$/$RCO_3$), and other radicals derived from alkanes
(ALKAP), alkenes (ALKEP), and isoprene (ISOP), which accounted for an additional
40% of the F(Ox). On a daytime basis, the maximum F(Ox) reached 7.4 ppb/h at
around 12:00 in the OCM period, while a persistent-high value (maximum of 12.5
ppb/h at 10:00–14:00) occurred in the LAM period. A vast amount of O$x$ was
consumed in the nitric acid (OH + $NO_2$) formation pathways, i.e., higher than the
ozonolysis removal. The daily averaged ozone production rates were 5.52 and 2.76
ppb/h during the LAM and OCM periods, respectively.



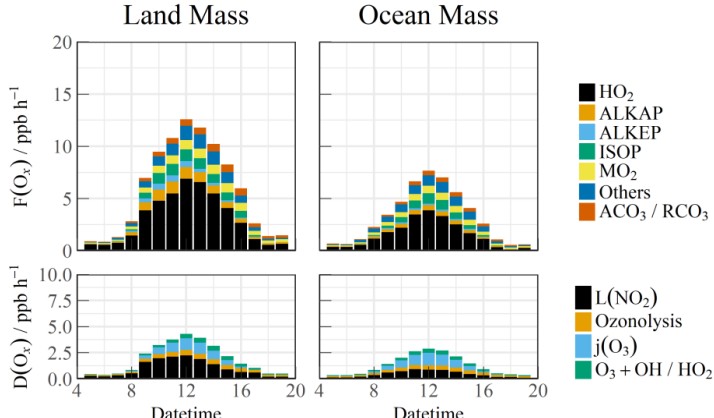


**Fig. 9.** The diurnal profiles of the speciation $F(O_X)$ and $D(O_X)$ during Land mass and Ocean mass episodes.
The data were calculated by the measured OH and $HO_2$ and modelled $RO_2$ radicals.

### 484 4.3 Relationship between precursors and oxidation rates

Despite the low level of human activities, oxidation precursors have an extended
lifetime in the stable atmosphere of coastal areas. Intensive photochemical reactions
occur after the accumulation of precursors, resulting in local net ozone production
comparable to that in the surrounding suburban environments (Zeren et al., 2022).
Simultaneous observations of both urban and coastal settings in Shenzhen have
indicated that the oxidation rates are comparable (Xia et al., 2021). The coupling of
precursor transport and local photochemical processes in marine areas makes it
meaningful to explore secondary pollution generation (Fig. 10(a), (b), and (c)). No
obvious radical source was missing during the LAM and OCM periods, and the
oxidation level was that expected from the base model. On a daytime basis, the mean
diurnal profile of the P(Ox) reached ~7 ppb/h in the LAM period, and the average
nitric acid (P($HNO_3$)) and sulfuric acid (P($H_2SO_4$)) production rates were ~1.6 and
~0.11 ppb/h, respectively. The P($HNO_3$) production rate was similar to the average of
observations in the Pearl River Delta region (~1.3 ppb/h), while that of the P($H_2SO_4$)
was only half the average level (~0.24 ppb/h) (Lu et al., 2013; Tan et al., 2019b; Yang
et al., 2022b). During the OCM period, the characteristics of the ocean air mass
alleviated the photochemical process, and the production rates of the secondary
pollutants decreased by approximately half and were close to the average levels in





winter (Ma et al., 2019).

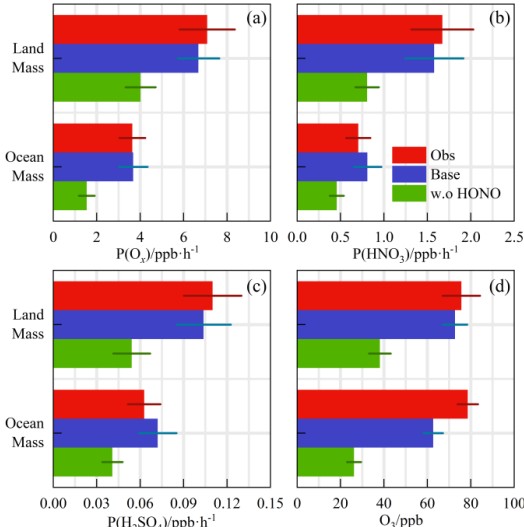

**Fig. 10.** The calculated reaction rates based on the observed concentrations for Land mass and Ocean mass episodes **(a)** P(Ox), **(b)** P(HNO₃), **(c)** P(H₂SO₄). **(d)** The observed and modelled O₃ concentration with a first-order loss term. The deposition process was equivalent to a lifetime of 15 hours to all species. All the rates and concentration are averaged for the daytime period between 10:00 and 15:00.

Contrary to numerous ocean observations, in the YMK site, intensive oxidation
was accompanied by a high diurnal HONO level (higher than 400 ppt) (Fig. 11). The
ozone levels were consistent with the Grade I air quality standard and far exceeded
the global background concentration (~40 ppb). Daytime photolysis reactions of
HONO contributed 1.52 ppb/h and 2.19 ppb/h to P(ROx) during the OCM and LAM
periods, respectively, which were much higher than the values in several megacities
during the photochemically polluted season (Tan et al., 2019a). Given the significance
of HONO photolysis in driving atmospheric chemistry, a sensitivity test was
conducted without constraints on HONO (i.e., w.o HONO) to specifically quantify the
contribution of HONO-induced secondary pollution. Only the homogeneous reaction
(OH + NO) participated in the formation of HONO in the default mode without
HONO input (Liu et al., 2022b). After evaluation, the P(Ox) was found to be 33% and
39% lower during the LAM and OCM periods, respectively, while the nitric acid and
sulfuric acid formation rates were 52% and 35% lower, respectively. The sensitivity
test identified the privileged role of the HONO-related mechanisms in the OH



chemistry, which resulted in a correlation between the efficient radical recycling and
secondary pollution.

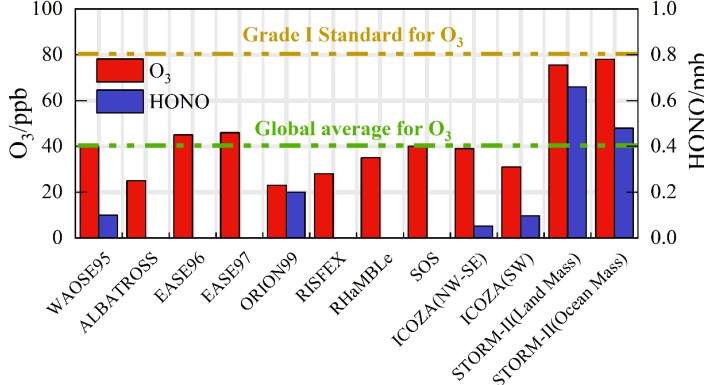

**Fig. 11.** Summary of both ozone and HONO concentrations in previous marine observations. The
concentrations are averaged for the daytime period between 10:00 and 15:00.
A time-dependent box model was used to test the association between the HONO
chemistry and the local ozone generation (Fig. 10(d)). In order to isolate the the $O_3$
photochemical production, the impacts of vertical entrainment and horizontal
advection were in general ignored. On the basis of the base scenario run, constraint of
the observed ozone concentration was removed, and the deposition process was
equivalent to a lifetime of 15 hours to all species. The observed and modelled $O_3$
concentrations in Fig. 10(d) are averaged for the daytime period between 10:00 and
15:00. The observed diurnal ozone concentrations were 75.7 ppb and 78.6 ppb during
the LAM and OCM periods, respectively. The daytime ozone was well reproduced by
the time-dependent box model, and the deviation of the simulation was less than 20%
(Fig. 10(d)). After removing the HONO constraint, the simulated ozone
concentrations were 38.2 and 26.3 ppb, i.e., 48% and 58% lower, during the LAM and
OCM periods, respectively. Both the HONO and ozone concentrations were reduced
to a low level (~70 ppt and ~35 ppb) and were close to several ocean observations
(Fig. 11) (Woodward-Massey et al., 2022b; Zhu et al., 2022; Xia et al., 2022). The
elevated daytime HONO had an additional effect on the oxidation in the background
atmosphere. For coastal cities, the particularity of the HONO chemistry in the MBL
tends to influence the ozone-sensitive system and eventually magnifies the ozone
background. Therefore, the promotion of oxidation by elevated precursor



concentrations is worth considering when formulating emission reduction policies.

# 5 Conclusions

Comprehensive observations of HOx radicals and other relevant species were
conducted in October 2019 at a coastal site in the Pearl River Delta (the YMK site,
22.55°N, 114.60°E). The overall air pollutants exhibited typical coastal features due to
the scarce anthropogenic emissions. The average daily maximum OH and $HO_2$
concentrations were $(4.7–9.5) \times 10^6$ cm$^{-3}$ and $(4.2–8.1) \times 10^8$ cm$^{-3}$, respectively. The
base RACM2-LIM1 model satisfactorily reproduced both the observed OH and $HO_2$
radical concentrations, but a slight overestimation of the OH radical occurred. The
daily maximum calculated total OH reactivity was 9.9 s$^{-1}$, and nearly 70% of the
reactivity was contributed by organic species.
In addition to anthropogenic and vegetation emissions, the synchronized air mass
transport from the northern cities and the South China Sea exerted a time-varying
influence on radical photochemistry and atmospheric oxidation. During the OCM
period, the observed OH and $HO_2$ radical concentrations could be reflected by the
base chemical mechanism, with daily average values of $4.5 \times 10^6$ cm$^{-3}$ and $4.9 \times 10^8$
cm$^{-3}$, respectively. The more active photochemical process during the LAM period
promoted the underestimation of the radical concentrations. Unmeasured reactive
species involved in oxidation propagation were responsible for elevated
photochemistry.
In the episode that was dominated by ocean mass, the $HO_2$ + NO reaction
accounted for ~50% of the primary OH yield. A higher OH generation rate was
found(12.6 ppb/h) during the LAM period, and the secondary source accounted for 67%
of the total, which was similar to several observations in polluted plumes. Reactions
between ROx and NOx and self-combination were the main pathways of radical
termination (~70%), and the contribution of peroxynitrite formation to the L(ROx)
could not be ignored in the daytime.
Intensive photochemical reactions occur after the accumulation of precursors,



resulting in local net ozone production comparable to that in the surrounding suburban environments. The daily average ozone production rates were 5.52 and 2.76 ppb/h in the LAM and OCM periods, respectively. The rapid oxidation process was accompanied by a higher diurnal HONO concentration (higher than 400 ppt). A non-HONO-constrained sensitivity test was performed to quantify the HONO-induced contribution to secondary pollution. After evaluation, the P(Ox) values were 33% and 39% lower during the LAM and OCM periods, respectively, while the nitric acid and sulfuric acid formation rates were 52% and 35% lower, respectively. The simulated daytime HONO and ozone concentrations were reduced to a low level (~70 ppt and ~35 ppb, respectively). For coastal cities, the particularity of the HONO chemistry in the MBL tends to influence the ozone-sensitive system and eventually magnifies the ozone background. Therefore, the promotion of oxidation by elevated precursor concentrations is worth considering when formulating emission reduction policies.

# Financial support

This work was supported by the National Natural Science Foundation of China (62275250, U19A2044, 61905003), the Natural Science Foundation of Anhui Province (No. 2008085J20), the National Key R&D Program of China (2022YFC3700301), and the Anhui Provincial Key R&D Program (2022l07020022).

# Data availability

The data used in this study are available from the corresponding author upon request (rzhu@aiofm.ac.cn).

# Author contributions

WQ Liu, PH Xie, RZ Hu contributed to the conception of this study. GX Zhang and RZ Hu performed the data analyses and manuscript writing. All authors contributed to measurements, discussed results, and commented on the paper.



## Competing interests

The contact author has declared that none of the authors has any competing interests.



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
