# Peer review of "Intensive photochemical oxidation in the marine atmosphere"

_EGUsphere, 2023_

## Author Comment (AC1)

Dear Editor and Referee,

Thanks for your suggestions which significantly help us to improve the manuscript. Hereby, we submit our responses and the manuscript has been revised accordingly. If there are any further questions or comments, please let us know.

Best regards

Renzhi Hu on behalf of all co-authors

Key Lab. of Environmental Optics & Technology, Anhui Institute of Optics and Fine Mechanics, Chinese Academy of Sciences

230031 Hefei China

E-mail: rzhu@aiofm.ac.cn

**Major Comments**

1. *In section 2, the authors should describe how the OCM and LAM sectors were assigned – by wind direction or trajectory analysis? Some of this description is provided in lines 261 – 270 and I suggest this is moved to section 2.*

**Reply:**

Thanks for your suggestion. The OCM and LAM sectors were assigned by trajectory analysis. We moved the detailed description to Section 2 (Line 132-140).

**Revision:**

Line 132-140: Using the hybrid single-particle Lagrangian integrated trajectory (HYSPLIT) model, the 24-h backward trajectories on special days were obtained. In Fig. S1, the red, blue, and green trajectories represent the results at altitudes of 100, 500, and 1000 m above ground level, respectively. Two typical transportation pathways dominated the air parcels. One originated from the northern megacities in the Pearl River Delta (defined as the land mass, LAM), especially on October 18, 19, and 27. In contrast, a clean air mass from the east or northeast was mainly transported to the observation site from the ocean (defined as the ocean mass, OCM), with representative episodes on October 22, 25, and 26.

2. *Section 2.1: some description of the vegetation type in the surrounding forest should be provided, so the reader can ascertain if other biogenic emissions such as monoterpenes were likely present that could influence the local chemistry.*

**Reply:**

Thanks for your suggestion. During the observation, the surrounding forest is lush around the YMK site. The vegetation type is evergreen broadleaf forests, which contributed to biogenic emissions. Previous literatures reported the monoterpene concentration in the YMK site, with a daily mean of 0.187 ppb (Zhu et al., 2021). Aboundant biogenic emissions will likely influence the local chemistry. We added the detailed description in Line 120-123.

**Revision:**

Line 120-123: Previous literatures reported the monoterpene concentration in the YMK site, with a daily mean of 0.187 ppb (Zhu et al., 2021). Aboundant biogenic emissions will likely influence the local chemistry.

**3. Line 132: '..via chemical transformation' add 'by addition of NO'. Please also state the purity of the NO and concentration of NO in the detection cell.**

**Reply:**

Thanks for your suggestion. The purity of the NO was mixed with 2% in $N_2$, and the concentration of NO in the detection cell was ~$1.6 \times 10^{12}$ cm$^{-3}$. NO was passed through a ferrous sulfate filter to remove impurities ($NO_2$, HONO, and so on) before being injected into the detection cell. We added the detailed description in Line 187-195.

**Revision:**

Line 187-195: For $HO_2$ measurement, the NO gas was mixed with 2% in $N_2$ to achieve $HO_2$-to-OH conversion. NO was passed through a ferrous sulfate filter to remove impurities ($NO_2$, HONO, and so on) before being injected into the detection cell. The NO concentration (~$1.6 \times 10^{12}$ cm$^{-3}$) corresponding to a conversion efficiency of ~15% was selected to avoid $RO_2 \rightarrow HO_2$ interference (especially from $RO_2$ radicals derived from long-chain alkanes ($C \geq 3$), alkenes, and aromatic hydrocarbons). Previous study denoted that the percentage interference from alkene-derived $RO_2$ under these operating conditions was no more than 5% (Wang et al., 2021).

**4. Line 147: Is this a single pass laser configuration or multi-pass? Please state**

**Reply:**

Thanks for your suggestion. The radical detection module utilized a single pass laser configuration, and the laser beam was amplified to a diameter of 8 mm. We stated the detailed description in Line 158-160.

**Revision:**

Line 158-160: The radical detection module utilized a single pass laser configuration,

and the laser beam was amplified to a diameter of 8 mm.

**5. *Line 155: change 'avoid' to 'reduce'**

**Reply:**

We followed the reviewer's comment. We changed the description in Line 167-170.

**Revision:**

Line 167-170: Efficient ambient air sampling was achieved using an aluminum nozzle (0.4 mm orifice), and the pressure in the chamber was maintained at 400 Pa via a vortex vacuum pump (XDS35i, Edwards) to reduce fluorescence quenching.

**6. *Line 159: The ozone interference as a function of ambient ozone and $H_2O$ (v) concentration and laser power determined from the laboratory experiment should be provided.**

**Reply:**

Thanks for your suggestion. We added the detailed description in Line 173-177.

**Revision:**

Line 173-177: Due to the synchronous reaction at 308nm, wavelength modulation is not applicable to ozone photolysis interference. Through laboratory experiments, at 20 mW laser energy, every 1% water vapor concentration and 50 ppb ozone concentration can generate a $2.5 \times 10^5$ cm$^{-3}$ OH concentration. The results in this paper have subtracted the ozone photolysis interference (Fig. S2).

[Figure]

**Fig. S2.** Mean diurnal profiles of measured [OH] before (red line) and after (blue line) deducting the O$_3$

interference. The coloured shadows denote the 25 and 75% percentiles. The grey areas denote nighttime.

7. *Line 160 -164: The previous good agreement reported for OH measurements by this system and the PKU-LIF in a previous study doesn't translate to an interference-free OH observation in the present study. Chemical removal of ambient OH using an inlet pre-injector is now seen as standard practice for LIF OH instruments. In the absence of this, the authors should provide some information on the chemical environment (ozone, alkene, NOx concentrations) were the instrument was deployed during the intercomparison and explain how this contrasts with the current environmental conditions.*

**Reply:**

Thanks for your suggestion. We will discuss whether internal interference exists in AIOFM-LIF from the following aspects:

First of all, literature research shows that measurement interference is more related to the length of the inlet in the low-pressure cell (Griffith et al., 2016). In terms of system design, the AIOFM-LIF system uses a short-length inlet design to minimize this and other unknown disturbances (The distance from radical sampling to fluorescence excitation is ~150 mm).

Additionally, potential interference may exist when the atmosphere contains abundant alkenes, ozone, and BVOCs, indicating that environmental conditions play leading roles in OH interferences (Mao et al., 2010; Fuchs et al., 2016; Novelli et al., 2014). An OH measurement comparison with a LIF instrument deployed an inlet pre-injector (PKU-LIF), was conducted in a real atmosphere in a previous study (Zhang et al., 2022). The ozonolysis interference on the measurement consistency of both systems was excluded under high-VOCs conditions. We have compared the chemical conditions during the intercomparison experiment and the current environmental conditions. Overall, the key parameters related to ozonolysis reactions ($O_3$、alkenes、isoprene and NOx) in YMK were similar to those during the comparison experiment, which is not conducive to generating potential OH interference. Therefore, it is not expected that OH measurement in the present study was affected by internal interference. We added

the detailed description in Line 177-187.

**Table.S1.** Comparison of key parameters related to ozonolysis reactions (O₃、 alkenes、 isoprene and NOx) between YMK and the intercomparison experiment. All the values are the diurnal average (10:00-15:00).

| Species | Intercomparison | YMK |
|---|---|---|
| $O_3$ (ppb) | 71.02 | 74.58 |
| Alkenes (ppb) | 1.29 | 1.10 |
| Isoprene (ppb) | 0.67 | 0.64 |
| NOx (ppb) | 5.65 | 4.24 |

**Revision:**

Line 177-187: In terms of system design, the AIOFM-LIF system incorporates a short-length inlet design to minimize interferences from ozonolysis and other unknown factors (the distance from radical sampling to flourescence excitation is ~150 mm). An OH measurement comparison with an interference-free instrument, PKU-LIF, was conducted in a real atmosphere in a previous study (Zhang et al., 2022). The ozonolysis interference on the measurement consistency of both systems was excluded under high-VOCs condition. Overall, the key parameters related to ozonolysis reactions (O₃、 alkenes、 isoprene and NOx) in YMK was similar to that during the intercomparison experiment, implies that the chemical conditions do not favor the generation of potential interference to OH measurement (Table S1).

8. *Line 166: The authors need to state the percentage interference from an alkene-derived RO2 under these operating conditions (it won't be zero).*

**Reply:**

Thanks for your suggestion. In the previous work, we have calculated the conversion efficiency of alkene-derived $RO_2$ to OH under different NO concentration (Wang et al., 2021). In the YMK observation, ethene accounted for about 70% of the total ethene concentration (Table S5). Therefore, we choose ethene and isoprene to investigate the percentage interference from an alkene-derived $RO_2$. When NO was at $1.6 \times 10^{12}$ cm$^{-3}$, the conversion efficiency of $HO_2$ was ~15%, and the percentage

interference from ethene and isoprene-derived $RO_2$ was 3.83% and 1.75%, respectively (Wang et al., 2021). We added the detailed description in Line 193-195.

**Revision:**

Line 193-195: Previous study denoted that the percentage interference from alkene-derived $RO_2$ under these operating conditions was no more than 5% (Wang et al., 2021).

**9. *Line 169: How much OH and HO2 was produced in the calibration? Was the calibration performed using a turbulent flow? How was the lamp flux determined? These details should be included in the manuscript.***

**Reply:**

Thanks for your suggestion. In the YMK observation, the calibration was performed in a laminar condition with a maximum flow rate of 20 SLM (standard liters per minute)(Wang et al., 2020; Wang et al., 2019). OH concentration was deduced by chemical radiometry according to the next Eq. :

$$[OH] = [HO_2] = \frac{1}{2} \cdot \frac{\sigma_{H_2O}}{\sigma_{O_2}} \cdot \frac{[H_2O]}{[O_2]} \cdot \frac{[O_3]}{P}$$

$\sigma_{H_2O}$ and $\sigma_{O_2}$ represent the absorption cross-sections of water and oxygen, respectively. $P$ represents the distribution factor of ozone concentration inside the tube. In previous studies, we obtained the actual measured values of $\sigma_{O_2}$ and $P$ through experiments (Wang et al., 2020). As the luminous flux in photolysis region is difficult to accurately measure, the linearly correlation between ozone concentration and 185 nm light flux was established. Ozone concentration in the flow tube was measured by a home-made Cavity Ring Down Spectrometer (CRDS, and the detection limit is 15 ppt@30 s, 1σ). Mercury lamp intensity is adjusted to establish.

In the YMK campaign, the humidity varied between 40 – 80% (Fig. S3). In order to test different atmospheric conditions, both low (~40%) and high (~70%) levels of water vapor were selected to produce OH and $HO_2$ radicals for calibration, and the corresponding HOx concentration obtained from the standard source was $1.0 \times 10^9$ cm$^{-3}$ and $1.8 \times 10^9$ cm$^{-3}$, respectively (Zhang et al., 2022).

We added the detailed description in Line 196-212.

**Revision:**

Line 196-212: A standard HOx radical source was used to complete the calibration of the detection sensitivity (Wang et al., 2020). The radical source is based on the simultaneous photolysis of $H_2O/O_2$ by a 185 nm mercury lamp. Humidified air flow is introduced to produce equal amounts of OH and $HO_2$ radicals after passing the photolysis region. The flow remained in a laminar condition with a maximum flow rate of 20 SLM (standard liters per minute). As the luminous flux in photolysis region is difficult to accurately measure, the linearly correlation between ozone concentration and 185 nm light flux was established. Ozone concentration in the flow tube was measured by a home-made Cavity Ring Down Spectrometer (CRDS, and the detection limit is 15 ppt@30 s, 1σ). Mercury lamp intensity is adjusted to establish. The instrument was calibrated every 1 or 2 days (except for shutdown during rainy periods), and the sensitivity used for the data processing was an average of all of the calibration results. In the YMK campaign, the humidity varied between 40 – 80% (Fig. S3). In order to test different atmospheric conditions, both low (~40%) and high (~70%) levels of water vapor were selected to produce OH and $HO_2$ radicals for calibration, and the corresponding HOx concentration obtained from the standard source was $1.0 \times 10^9 \, \text{cm}^{-3}$ and $1.8 \times 10^9 \, \text{cm}^{-3}$, respectively (Zhang et al., 2022).

*10. Section 2.2.2: Given the importance of HONO as an OH source in this study, some comments on possible instrument artefacts/or how interferences were corrected for should be discussed.*

**Reply:**

Thank you for your suggestion. HONO measurement was conducted using a commercial Long-Path Absorption Photometer (LOPAP). The LOPAP method utilizes two absorption tubes in series for differential correction, which effectively eliminates the influence of known interfering substances such as $NO_2$ and $N_2O_5$, offering an advantage over traditional wet chemistry methods. This method has been extensively tested for its suitability in detecting HONO in complex atmospheric conditions, as

demonstrated in previous studies by(Yang et al., 2022a; Yang et al., 2021b; Wang et al., 2023). During the YMK campaign, zero air measurements were taken every 8 hours for a duration of 20 minutes to correct for instrument baseline fluctuations. Additionally, a liquid nitrite standard calibration was performed on a weekly basis to ensure the accuracy of the calibration curve used for measuring HONO concentrations. We added the detailed description in Line 225-233.

**Revision:**

Line 225-233: HONO measurement was conducted using a commercial Long-Path Absorption Photometer (LOPAP). The LOPAP method utilizes two absorption tubes in series for differential correction, which effectively eliminates the influence of known interfering substances such as $NO_2$ and $N_2O_5$, offering an advantage over traditional wet chemistry methods. Zero air measurements were taken every 8 hours for a duration of 20 minutes to correct for instrument baseline fluctuations. This method has been extensively tested for its suitability in detecting HONO in complex atmospheric conditions, as demonstrated in previous studies by (Yang et al., 2022a; Yang et al., 2021b; Wang et al., 2023).

*11. Line 183: Which of the measured photolysis rates were used as model constraints?*

**Reply:**

Thanks for your suggestion. Eight measured photolysis rates ($j(NO_2)$, $j(H_2O_2)$, $j(HCHO)$, $j(HONO)$, $j(NO_2)$, $j(NO_3)$, $j(O^1D)$) were used as model constraints.

**Revision:**

Line 233-234: Eight measured photolysis rates ($j(NO_2)$, $j(H_2O_2)$, $j(HCHO)$, $j(HONO)$, $j(NO_2)$, $j(NO_3)$, $j(O^1D)$) were used as model constraints.

*12. Line 204: Did the modelled OH and HO2 reach a steady state concentration during this time – what was the % difference between day 2 and day 3 radical concentrations?*

**Reply:**

We followed the reviewer's comment. We calculated the steady state concentrations of OH and HO$_2$ radical using the Eq.(1)(2):

$$[OH]_{PSS} = \frac{j_{HONO}[HONO] + \varphi_{OH}j(O^1D)[O_3] + k_{HO_2+NO}[NO][HO_2]}{k_{OH}} \tag{1}$$

$$[HO_2]_{PSS} = \frac{k_{CO+OH}[CO][OH] + j_{HCHO}[HCHO] + k_{RO_2+NO}[NO][RO_2]}{k_{HO_2+NO}[NO]} \tag{2}$$

Due to the lack of RO$_2$ radical observation data, substitute the $[RO_2]$ and $k_{OH}$ items in Eq.(1)(2) with the simulated value. The comparison of steady state and the modelled concentrations in the daytime were shown in Fig.3. During the entire observation period, the base model reached a steady state and showed good agreement with the calculated concentrations of OH and HO$_2$ radicals using Eq.(1)(2). Specifically, on the second and third days, there were no significant differences between the steady-state calculation and the base model, with OH and HO2 concentration deviations of 7.5% and 3.1%, respectively.

[Figure]

**Fig. 3.** Timeseries of the observed and modelled parameters for OH, HO$_2$ and $k_{OH}$ during the observation period. **(a)** OH, **(b)** HO$_2$, **(c)** $k_{OH}$.

**Revision:**

Line 262-267: In addition, another steady-state calculation method (PSS) can also be used to estimate the concentrations of OH and HO$_2$ radicals (Eq. (1)(2), (Woodward-Massey et al., 2022; Slater et al., 2020)). Since the $k_{OH}$ and RO$_2$ concentrations were not obtained in this observation, simulated values are used as substitutes. Other radical and reactive intermediates are actual values that measured from the instruments in Table S2.

Line 369-374: Overall, the observed OH and HO$_2$ concentrations were both well reproduced by the base model incorporating the RACM2-LIM1 mechanism. The observed OH was underestimated only on the first days, and a slight model overestimation happened on October 23&24. PSS calculation showed good agreement with the base model, providing evidence of the balance of radical internal consistency in the daytime.

**13. Line 208: Could the authors explain their choice of the 18 hr lifetime? Were any model tests performed to assess how well the model predicted HCHO for example (if left unconstrained to HCHO) with an 18 hr lifetime? How sensitive were the modelled OH, HO2 and modelled kOH to the choice of this lifetime?**

**Reply:**

Thanks for your suggestion. After literature research, we found that the lifetime for the model-generated intermediate is usually set between 8 – 24 h (Ma et al., 2022; Yang et al., 2021a; Whalley et al., 2021). We tested the relationship between the first-order loss term and simulated OH, HO$_2$, $k_{OH}$ by changeing the lifetime (8h, 12h, 18h, and 24h). The results have been added to Fig. S4. The sensitivity analysis shows that when the lifetime changes within 8 – 24 hours (8h, 12h, 18h, and 24h). The values differed less than 5% between two cases for both OH, HO$_2$, $k_{OH}$. Therefore, we finally chose a settling time of 18 hours, corresponding to first order loss rate of ~1.5 cm/s (by assuming a boundary layer height of about 1 km).

[Figure]

**Fig. S4.** The relationship between the first-order loss term and simulated **(a)** OH, **(b)** HO$_2$, **(c)** $k_{OH}$ by changeing the lifetime within 8 – 24 hours (8h, 12h, 18h, and 24h).

To test the sensitivity of the HCHO simulation, data from October 18th to 22nd were selected (in the Figure below). The study found that when the lifetime was altered

between 8 and 24 hours, the simulated HCHO concentrations changed in a manner consistent with the observed values from the YMK site. However, the model tended to overestimate formaldehyde concentrations over longer time periods. This phenomenon of overestimation has been found in other areas as well (Li et al., 2014).

[Figure]

We added the detailed description in Line 255-260.

**Revision:**

Line 255-260: The physical losses of species due to processes such as deposition, convection, and advection were approximately replaced by an 18 h atmospheric lifetime, corresponding to first order loss rate of ~1.5 cm/s (by assuming a boundary layer height of about 1 km). The sensitivity analysis shows that when the lifetime changes within 8 – 24 hours, the values differed less than 5% for both OH, HO$_2$, $k_{OH}$ (Fig. S4).

*14. Line 211 – 217: What was the modelled BrO concentration when the model was run with Br2 chemistry? I think some comment on the potential impact of iodine chemistry should be included in this section (as other papers which focus on the chemistry of the marine boundary layer consider both iodine and bromine chemistry). With regards to the Tables S2 and S3 provided in the SI, what are the expected products from the photolysis reactions in S2? HOBr photolysis is a source of OH, was this included as the photolysis product in the model? For S3, 'ACD' and 'MO2' need defining. Were heterogeneous loss processes for HOBr considered? In Bloss et al., ACP, 2010, the reaction of CH3O2 + BrO produces HOBr + CH2O2 (which dissociates to CO and H2O). Could the authors*

*explain/provide a reference for their choice of products (HOBr, HO2 and HCHO) from this reaction? As it is written, the reactants and products don't balance. As it stands, the halogen scheme included seems incomplete and I suggest this is reviewed before final publication.*

**Reply:**

In the previous manuscript, when the model was run with $Br_2$ chemistry, the diurnal concentration of BrO was depicted in the following Figure. During the observation period, BrO concentration exhibited a clear diurnal variation with peak concentrations at 0.68 ppt. This value is consistent with the simulated results observed by HZ (~0.5 ppt) but lower than those obtained at CHABLIS (~5.0 ppt) (Bloss et al., 2010; Xia et al., 2022).

[Figure]

Some deficiencies of Tables S3 and S4 were identified in the previous manuscript version, which have now been addressed. Iodine-related mechanisms are also considered, and the photolysis products in Table S3 have been added, including the photolysis of HOBr (the uptake process of HOBr was not considered). ACD and $ACO_3$ represent Acetaldehyde and Acetyl peroxy radicals, respectively, in the RACM2 mechanism. Meanwhile, $MO_2$ represents Methyl peroxy radicals. Explanations for ACD, $ACO_3$ and $MO_2$ have also been added to the table notes in Table S4. Additionally, we agree with the authors' suggestion that the subsequent decomposition products of $CH_3O_2 + BrO$ actually yield $HOBr + CO + H_2O$ instead of $HOBr + HO_2 + HCHO$.

In order to better explore the effect of Br and I chemistry on HOx radicals, we

chose BrO/IO as the initiation point of halogen chemistry in the latest version of the manuscript. The concentration of BrO and IO is set to ~5 ppt, which is a typical level in MBL site (Xia et al., 2022; Bloss et al., 2010; Whalley et al., 2010). In this scenario (Fig. 4, green line). The daytime concentration of $HO_2$ radical decreased by 8.5% and 13.3% during the LAM and OCM periods, respectively, compared to the base model. However, there was no significant change in the concentration of OH radicals (<3%).

[Figure]

**Fig. 4.** Median diurnal profiles of the observed and modelled OH, $HO_2$, $k_{OH}$ during LAM and OCM episodes. The coloured shadows for OH and $HO_2$ radicals denote the 25 and 75% percentiles. The grey areas denote nighttime.

We added the detailed description in Line 417-426.

**Revision:**

Line 417-426: Halogen species have been recognized as potent oxidizers that can boost photochemistry (Xia et al., 2022; Peng et al., 2021). A sensitivity test was performed by imposing BrO and IO into the base model to diagnose the impact of the halogen chemistry on the troposphere chemistry. The concentration of BrO and IO is set to ~5 ppt, which is a typical level in MBL site (Xia et al., 2022; Bloss et al., 2010; Whalley et al., 2010). The details of the mechanisms involved are listed in Tables S3 and S4. In this scenario (Fig. 4, green line). The daytime concentration of $HO_2$ radical decreased by 8.5% and 13.3% during the LAM and OCM periods, respectively, compared to the

base model. However, there was no significant change in the concentration of OH radicals (<3%).

Table S3-S4:

**Table.S3**. Photolysis frequencies for Br-related species (Atkinson et al., 2007; Bloss et al., 2010).

| Reaction | Mean $j(x)$ / $j(NO_2)$ | References |
|---|---|---|
| $Br_2 + hv$ --> Br+Br | 3.45 | (Atkinson et al., 2007) |
| $BrO + hv$ --> Br+O | 5.41 | (Atkinson et al., 2007) |
| $BrONO_2$ --> $Br + NO_3$ | 0.16 | (Atkinson et al., 2007) |
| $BrONO + hv$ --> $Br + NO_2$ | 1.14 | (Atkinson et al., 2007) |
| $BrONO + hv$ --> $BrO + NO$ | 1.14 | (Atkinson et al., 2007) |
| $HOBr + hv$ --> Br + OH | 0.256 | (Atkinson et al., 2007) |
| $I_2 + hv$ --> I + I | 20.30 | (Atkinson et al., 2007) |
| $IO + hv$ --> I + O | 18.30 | (Bloss et al., 2001) |
| $OIO + hv$ --> $I + O_2$ | 2.58 | (Cox et al., 1999) |
| $IONO_2 + hv$ --> $I + NO_3$ | 0.556 | (Joseph et al., 2007) |
| $I_2O_2 + hv$ --> IO + IO | 0.556 | (Joseph et al., 2007) |
| $I_2O_3 + hv$ --> IO + OIO | 0.556 | (Joseph et al., 2007) |
| $I_2O_4 + hv$ --> OIO + OIO | 0.556 | (Joseph et al., 2007) |
| $INO_2 + hv$ --> $I + NO_2$ | 0.319 | (Bloss et al., 2010) |
| $INO + hv$ --> I + NO | 3.71 | (Bloss et al., 2010) |
| $HOI + hv$ --> OH + I | 1.12 | (Atkinson et al., 2007) |

**Table.S4**. Gas-phase kinetics for Br-related species in RACM2 mechanism. Revised by (Bloss et al., 2010). ACD and $ACO_3$ represent Acetaldehyde and Acetyl peroxy radicals, respectively, in the RACM2 mechanism. Meanwhile, $MO_2$ represents Methyl peroxy radicals. $PI_1$, $PI_2$, $PI_3$, $PI_4$ are the particulate iodine.

| Reaction | Reaction rate constant ($cm^3s^{-1}$) | References |
|---|---|---|
| $Br + O_3$ --> $BrO + O_2$ | $1.7 \times 10^{-11}exp(-800/T)$ | (Atkinson et al., 2007) |
| $Br + HO_2$ --> $HBr + O_2$ | $7.7 \times 10^{-12}exp(-450/T)$ | (Atkinson et al., 2007) |
| $HBr + OH$ --> $Br + H_2O$ | $6.7 \times 10^{-12}exp(155/T)$ | (Atkinson et al., 2007) |
| $Br_2 + OH$ --> HOBr + Br | $2.0 \times 10^{-11}exp(240/T)$ | (Atkinson et al., 2007) |
| $Br + HCHO$ --> HBr + HCO | $7.7 \times 10^{-12}exp(-580/T)$ | (Atkinson et al., 2007) |
| $Br + ACD$ --> $HBr + ACO_3$ | $1.8 \times 10^{-11}exp(-460/T)$ | (Atkinson et al., 2007) |
| $Br + NO_2$ --> BrONO | $k_0 = 4.2 \times 10^{-31}exp(T/300)^{-2.4}$ $k_\infty = 2.7 \times 10^{-11}$, $F_c = 0.6$ | (Bloss et al., 2010) |
| $BrO + BrO$ --> $Br + Br + O_2$ | $2.7 \times 10^{-12}$ | (Atkinson et al., 2007) |
| $BrO + BrO$ --> $Br_2 + O_2$ | $2.9 \times 10^{-14}exp(840/T)$ | (Atkinson et al., 2007) |
| $BrO + HO_2$ --> $HOBr + O_2$ | $4.5 \times 10^{-12}exp(500/T)$ | (Atkinson et al., 2007) |
| $HO + HOBr$ --> $BrO + H_2O$ | $5.0 \times 10^{-11}$ | (Bloss et al., 2010) |
| $BrO + MO_2$ --> $HOBr + CO + H_2O$ | $4.6 \times 10^{-13}exp(798/T)$ | (Enami et al., 2007) |
| $BrO + NO$ --> $Br + NO_2$ | $8.7 \times 10^{-12}exp(260/T)$ | (Atkinson et al., 2007) |
| $BrO + NO_2$ --> $BrONO_2$ | $k_0 = 5.2 \times 10^{-31}exp(T/300)^{-3.2}$ $k_\infty = 6.9 \times 10^{-12}exp(T/300)^{-2.9}$, $F_c = 0.6$ | (Bloss et al., 2010) |
| $BrONO_2$ --> $BrO + NO_2$ | $2.8 \times 10^{13}exp(12360/T)$ | (Orlando and Tyndall, 1996) |
| $I + O_3$ --> $IO + O_2$ | $k = 2.1 \times 10^{-11}exp(-830/T)$ | (Atkinson et al., 2007) |
| $I + HO_2$ --> $HI + O_2$ | $k = 1.5 \times 10^{-11}exp(-1090/T)$ | (Atkinson et al., 2007) |
| $OH + HI$ --> $I + H_2O$ | $k = 1.6 \times 10^{-11}exp(440/T)$ | (Atkinson et al., 2007) |
| $OH + I_2$ --> HOI + I | $k = 2.1 \times 10^{-10}$ | (Atkinson et al., 2007) |
| $NO_3 + I_2$ --> $I + IONO_2$ | $k = 1.5 \times 10^{-12}$ | (Atkinson et al., 2007) |

| | | |
|---|---|---|
| $NO_3 + HI \rightarrow HNO_3 + I$ | $k = 1.3 \times 10^{-12}\exp(-1830/T)$ | (Atkinson et al., 2007) |
| $I + NO_2 \rightarrow INO_2$ | $k_0 = 3.0 \times 10^{-31}(T/300)^{-1.0}$
 $k_\infty = 6.6 \times 10^{-11}, \quad F_c = 0.6$ | (Bloss et al., 2010) |
| $INO_2 \rightarrow I + NO_2$ | $k = 0.14 \text{ s}^{-1}$ (at 268 K) | (Bloss et al., 2010) |
| $INO_2 + INO_2 \rightarrow I_2 + 2NO_2$ | $k = 4.7 \times 10^{-13}\exp(-1670/T)$ | (Atkinson et al., 2007) |
| $I + NO \rightarrow INO$ | $k_0 = 1.8 \times 10^{-32}(T/300)^{-1.0}$
 $k_\infty = 1.7 \times 10^{-11}, \quad F_c = 0.6$ | (Bloss et al., 2010) |
| $INO \rightarrow I + NO$ | $k = 0.087 \text{ s}^{-1}$ (at 268 K) | (Bloss et al., 2010) |
| $INO + INO \rightarrow I_2 + NO + NO$ | $k = 8.4 \times 10^{-11}\exp(-2620/T)$ | (Atkinson et al., 2007) |
| $IO + IO \rightarrow 2I + O_2$ | $k = 0.11 \times 5.4 \times 10^{-11}\exp(180/T)$ | (Atkinson et al., 2007) |
| $IO + IO \rightarrow I + OIO$ | $k = 0.38 \times 5.4 \times 10^{-11}\exp(180/T)$ | (Atkinson et al., 2007) |
| $IO + IO \rightarrow I_2O_2$ | $k = 0.51 \times 5.4 \times 10^{-11}\exp(180/T)$ | (Atkinson et al., 2007) |
| $IO + HO_2 \rightarrow HOI + O_2$ | $k = 1.4 \times 10^{-11}\exp(540/T)$ | (Atkinson et al., 2007) |
| $OH + HOI \rightarrow IO + H_2O$ | $k = 1.0 \times 10^{-10}$ | (Dillon et al., 2006) |
| $IO + CH_3O_2 \rightarrow CH_3O + IOO$ | $k = 2.0 \times 10^{-12}$ | (Dillon et al., 2006) |
| $IO + NO \rightarrow I + NO_2$ | $k = 7.15 \times 10^{-12}\exp(300/T)$ | (Atkinson et al., 2007) |
| $IO + NO_2 \rightarrow IONO_2$ | $k_0 = 6.5 \times 10^{-31}(T/300)^{-3.5}$
 $k_\infty = 7.6 \times 10^{-12}(T/300)^{-1.5}, F_c = 0.6$ | (Bloss et al., 2010) |
| $IONO_2 \rightarrow IO + NO_2$ | $k = 2.1 \times 10^{15}\exp(-13670/T)$ | (Kaltsoyannis and Plane, 2008) |
| $IO + NO_3 \rightarrow OIO + NO_2$ | $k = 9.0 \times 10^{-12}$ | (Dillon et al., 2006) |
| $I + NO_3 \rightarrow IO + NO_2$ | $k = 1.0 \times 10^{-12}$ | (Dillon et al., 2006) |
| $IO + BrO \rightarrow I + Br + O_2$ | $k = 0.2 \times 1.5 \times 10^{-11}\exp(510/T)$ | (Atkinson et al., 2007) |
| $IO + BrO \rightarrow Br + OIO$ | $k = 0.8 \times 1.5 \times 10^{-11}\exp(510/T)$ | (Atkinson et al., 2007) |
| $IO + OIO \rightarrow I_2O_3$ | $k = 5 \times 10^{-11}$ | (Martin et al., 2009) |
| $OIO + OIO \rightarrow I_2O_4$ | $k = 1.5 \times 10^{-10}$ | (Martin et al., 2009) |
| $OIO + I_2O_3 \rightarrow PI_1$ | $k = 1.5 \times 10^{-10}$ | (Martin et al., 2009) |
| $OIO + I_2O_4 \rightarrow PI_2$ | $k = 1.5 \times 10^{-10}$ | (Martin et al., 2009) |
| $I_2O_2 + O_3 \rightarrow I_2O_3 + O_2$ | $k = 1.0 \times 10^{-12}$ | (Saunders and Plane, 2005) |
| $I_2O_3 + O_3 \rightarrow I_2O_4 + O_2$ | $k = 1.0 \times 10^{-12}$ | (Saunders and Plane, 2005) |
| $I_2O_4 + O_3 \rightarrow PI_3$ | $k = 1.0 \times 10^{-12}$ | (Saunders and Plane, 2005) |
| $I_2O_2 \rightarrow IO + IO$ | $k = 10.0 \text{ s}^{-1}$ | (Kaltsoyannis and Plane, 2008) |
| $I_2O_4 \rightarrow OIO + OIO$ | $k = 0.1 \text{ s}^{-1}$ | (Kaltsoyannis and Plane, 2008) |
| $NO + OIO \rightarrow IO + NO_2$ | $k = 1.1 \times 10^{-12}\exp(542/T)$ | (Plane et al., 2006) |
| $OH + OIO \rightarrow PI_4(HIO_3)$ | $k = 2.2 \times 10^{-10}\exp(243/T)$ | (Plane et al., 2006) |
| $BrO + DMS \rightarrow Br + DMSO$ | $k = 1.4 \times 10^{-14}\exp(950/T)$ | (Bloss et al., 2010) |
| $Br + DMS \rightarrow HBr + CH_3SCH_2$ | $k = 9.0 \times 10^{-11}\exp(-2390/T)$ | (Bloss et al., 2010) |
| $IO + DMS \rightarrow I + DMSO$ | $k = 1.2 \times 10^{-14}$ | (Bloss et al., 2010) |

*15. Line 233: 'exhibited good consistency..' it would be useful to provide typical concentrations of CO and PM2.5 to aid comparison to the previous campaigns referenced.*

**Reply:**

Thanks for your suggestion. We have added the range of CO and PM$_{2.5}$ concentrations in Line 326-328.

**Revision:**

Line 326-328: The CO and PM$_{2.5}$ concentrations exhibited good consistency and even mild pollution features ((0.36 ± 0.12 ppm) and (37.70 ± 7.91 μg/m$^3$), respectively),

reflecting the influence of human activities.

**16. Section 3.1.1: As the paper is trying to contrast LAM and OCM sectors, I think it would be useful from the start of this section to provide concentrations for the species discussed (e.g. NMHCs, NOx, CO, O3) from both sectors rather than campaign averages.**

**Reply:**

Thanks for your suggestion. We have modified the Section to provide concentrations for the species discussed (e.g. NMHCs, NOx, CO, $O_3$) from both sectors rather than campaign averages.

**Revision:**

Line 301-351:

As typical marine air components, the concentrations of NOx, CO, $PM_{2.5}$, and other pollutants were lower than those detected in other observation campaigns in both urban and suburban areas in the Pearl River Delta region (Tan et al., 2019; Lu et al., 2012; Yang et al., 2022b). Serval observation campaigns have discovered the relationship between wind direction and radical chemistry (Lu et al., 2012; Fuchs et al., 2017; Niu et al., 2022). Although there was no apparent wind speed condition, the dominant air mass still influenced the pollutant concentrations due to the particularity of the marine site.

During the OCM period, the NOx and HCHO concentrations exhibited relatively clean characteristics that were consistent with those previously observations in open ocean (RHaMBLe, SOS, CHABLIS and ALBATROSS, Table 1). Isoprene, a representative BVOC, achieved a diurnal concentration of $0.58 \pm 0.06$ ppb, indicated slightly local emissions could have impacted the concentrations of the precursor species even in OCM sector. The ozone concentration in the YMK site was always at the critical value of the updated Class I standard (GB3095-2012, average hourly $O_3$ of 81 ppb at 25°C and 1013 kPa). The occurrence of fewer emissions reduced the titration effect, resulting in the ozone exhibiting no apparent diurnal trend on some of the dates and a

high background value at night (78.1 ± 7.6 ppb).

As a coastal site, chemical conditions could be influenced by local land emissions depending on the wind direction. Compared with the OCM period, the meteorological conditions (T, RH, and J-values) changed slightly during the LAM episode, but the pollutants were accumulated due to the transport of the plume from the northern cities (Fig. 2). The CO and $PM_{2.5}$ concentrations exhibited good consistency and even mild pollution features ((0.36 ± 0.12 ppm) and (37.70 ± 7.91 μg/m$^3$), respectively), reflecting the influence of human activities. Both NO and $NO_2$ peaked at around 10:00, exhibiting prominent pollution characteristics. HONO exhibited a distribution with high daytime (0.66 ± 0.08 ppb) and low nighttime (0.33 ± 0.09 ppb) concentrations. This unique distribution of HONO has been observed in remote environments in several previous observation campaigns (Jiang et al., 2022; Crilley et al., 2021). High HONO concentration in the daytime will affect the chemical composition of the atmosphere and the secondary pollution generation.

The detailed information for VOCs species during the YMK campaign has been added in the Table S5. The daily maximum NMHC concentration peaked at 27.81 ± 9.91 ppb, and the maximum value of ~40 ppb occurred on October 27. Local biological emissions significantly affected the NMHC composition of the site, and isoprene achieved a noon maximum of 0.82 ± 0.16 ppb. Neither anthropogenic alkenes (2.21 ± 0.94 ppb) nor aromatic (1.31 ± 0.25 ppb) hydrocarbons were abundant, and OVOCs accounted for approximately 50% of the total. As a photochemical indicator, formaldehyde peaked at ~4 to ~8 ppb during the LAM episode, suggesting a more vigorous oxidation process. The HONO concentration was 6.8 times higher than the SW scenario in the ICOZA observation (a pollution period dominated by a southwest wind direction), while the HCHO concentration was 3.1 times higher. (Woodward-Massey et al., 2022). The abundance of oxidation precursors (HONO, HCHO, $O_3$, and NMHCs) reflected the unique atmospheric conditions in the marine environment in China, which originated from the complex atmospheric pollution.

*17. Line 235: I don't think it is 'conventional belief' that marine ozone would*

*necessarily be at background levels. At coastal sites which are influenced by land emissions, as is the case at the YMK site, I don't think it is unexpected to observe net ozone production given the NOx concentrations reported. I think it would be valuable to highlight, perhaps in Table 1, contrasting marine environments – for example, some of the referenced literature are from marine sites which are considered representative of the open ocean (RHaMBLe, SOS, ALBATROSS), whereas others, including YMK, are coastal sites which, depending on the wind direction, could be influenced by local land emissions.*

**Reply:**

Thanks for your suggestion. We adopted the suggestions of reviewers and marked the types of ocean observations in Table1. Three of the observations by RHaMBLe, SOS, and ALBATROSS were classified as open ocean, while the others were considered coastal features. We have also made some changes in the manuscript (Line 312-314&322-323).

**Revision:**

Line 312-314: During the OCM period, the NOx and HCHO concentrations exhibited relatively clean characteristics that were consistent with those previously observations in open ocean (RHaMBLe, SOS, CHABLIS and ALBATROSS, Table 1).

Line 322-323: As a coastal site, chemical conditions could be influenced by local land emissions depending on the wind direction.

*18. Line 246: provide typical concentrations of alkenes and aromatics.*

**Reply:**

Thanks for your suggestion. We have added the typical concentrations of alkenes and aromatics during the daytime (10:00 – 15:00). The detailed information table for VOCs species during the YMK campaign has been added in the Table S5.

**Revision:**

Line 342-344: Neither anthropogenic alkenes (2.21 ± 0.94 ppb) nor aromatic (1.31 ± 0.25 ppb) hydrocarbons were abundant, and OVOCs accounted for approximately 50%

of the total.

Table. S5. The detailed information table for VOCs species during the YMK campaign. The mean concentration, standard deviation (SD), minimum value (Min), maximum value (Max), and percentage contribution in the species for the top-five ranked species in alkanes, alkenes, aromatic and OVOCs are listed. All the values are the daily average (0:00-24:00).

| Species | Mean (ppb) | Sd (ppb) | Min (ppb) | Max (ppb) | Proportion (%) |
|---|---|---|---|---|---|
| Alkane | | | | | |
| ethane | 1.72 | 0.564 | 0.24 | 5.621 | 29.2 |
| propane | 1.246 | 0.524 | 0.136 | 5.438 | 21.15 |
| n-butane | 0.646 | 0.395 | 0.054 | 2.424 | 10.97 |
| i-butane | 0.561 | 0.471 | 0.029 | 3.372 | 9.52 |
| n-hexane | 0.41 | 0.307 | 0.033 | 3.026 | 6.96 |
| Alkene | | | | | |
| ethene | 0.592 | 0.656 | 0.034 | 5.48 | 69.08 |
| propene | 0.123 | 0.127 | 0.017 | 1.187 | 14.35 |
| 1-butene | 0.046 | 0.014 | 0.012 | 0.107 | 5.37 |
| trans-2-butene | 0.028 | 0.006 | 0.006 | 0.05 | 3.27 |
| cis-2-butene | 0.026 | 0.006 | 0.007 | 0.045 | 3.03 |
| Aromatic | | | | | |
| toluene | 0.523 | 0.361 | 0.035 | 2.82 | 38.34 |
| benzene | 0.286 | 0.112 | 0.032 | 0.742 | 20.97 |
| m-xylene | 0.123 | 0.237 | 0.015 | 3.579 | 9.02 |
| ethyl benzene | 0.107 | 0.134 | 0.017 | 2.052 | 7.84 |
| o-xylene | 0.103 | 0.214 | 0.015 | 3.294 | 7.55 |
| OVOC | | | | | |
| acetone | 3.297 | 0.835 | 0.412 | 5.978 | 52.47 |
| acetaldehyde | 1.742 | 0.635 | 0.276 | 5.805 | 27.73 |
| methyl ethyl ketone | 0.496 | 0.15 | 0.051 | 1.118 | 7.89 |
| methyl t-butyl ether | 0.213 | 0.208 | 0.018 | 1.512 | 3.39 |
| propionaldehyde | 0.178 | 0.081 | 0.028 | 0.572 | 2.83 |

*19. Line 260: The wind speed during the campaign is low, so I would expect local emissions could have impacted the concentrations of the precursor species to a certain extent. 0.5 ppb isoprene was observed in the OCM sector (fig.2) which, to me, suggests some local influences which should be acknowledged.*

**Reply:**

Thanks for your suggestion. We agree with the reviewer that YMK site is also

partially affected by emissions even during the OCM period. We acknowledge this in revised manuscript (Line 314-317).

**Revision:**

Line 314-317: Isoprene, a representative BVOC, achieved a diurnal concentration of 0.58 ± 0.06 ppb, indicated slightly local emissions could have impacted the concentrations of the precursor species even in OCM sector.

*20. Line 319: In previous literature, e.g. Whalley et al., ACP, 2010, the inclusion of halogen chemistry led to an increase in modelled OH concentrations and a decrease in modelled HO2 concentration, so the decrease in the modelled OH concentration reported here is a little surprising – perhaps the differing levels of NOx between this study and RHaMBLe play a role? Could the authors provide a little more detail on the dominant reactions in the halogen scheme that are contributing to OH destruction?*

**Reply:**

[Figure]

**Fig. 4.** Median diurnal profiles of the observed and modelled OH, HO$_2$, $k_{OH}$ during LAM and OCM episodes. The coloured shadows for OH and HO$_2$ radicals denote the 25 and 75% percentiles. The grey areas denote nighttime.

Thanks for your suggestion. In order to better explore the effect of Br and I

chemistry on HOx radicals, we chose BrO/IO as the initiation point of halogen chemistry in the latest version of the manuscript. The concentration of BrO and IO is set to ~5 ppt, which is a typical level in MBL site (Xia et al., 2022; Bloss et al., 2010; Whalley et al., 2010). In this scenario (Fig. 4, green line). The daytime concentration of $HO_2$ radical decreased by 8.5% and 13.3% during the LAM and OCM periods, respectively, compared to the base model. However, there was no significant change in the concentration of OH radicals (<3%).

[Figure]

**Fig. S6.** By modifying the NO concentration in different levels (Scenario 1: [NO]×150%, Scenario 2: base, Scenario 3: [NO]×20%, Scenario 4: [NO]×10%), the response of HOx radicals to the halogen mechanism varied under different NO levels (30 – 500 ppt in the diurnal time).

Traditionally, it is believed that the inclusion of halogen chemistry leads to higher modeled OH concentrations and lower modeled $HO_2$ concentrations. Therefore, the lack of an increase in OH concentration with the introduction of the halogen mechanism at the YMK site calls for further investigation (Fig. S6). By modifying the NO concentration in different levels (Scenario 1: [NO]×150%, Scenario 2: base, Scenario 3: [NO]×20%, Scenario 4: [NO]×10%), the response of HOx radicals to the halogen

mechanism varied under different NO levels.

As the constrained NO increased from 30 ppt to 500 ppt, the reduction in $HO_2$ radicals due to the Br and I mechanisms ranged between 10% and 20%. At elevated NOx levels, reactions between halogen radicals and NOx occurred, inhibiting the formation of OH radicals. In Scenario 1, the OH concentration even decreased by 3.5% when introducing the halogen mechanism. When NO concentration was constrained around 30 ppt (Scenario 4), similar to those obtained in RHaMBLe/CYPHEX campaigns, the modelled OH concentration increased by 14.4%, while the $HO_2$ concentration decreased by approximately 20.8% (Whalley et al., 2010; Bloss et al., 2010). Therefore, the sensitivity of OH radicals to the halogen mechanism in the YMK region is primarily limited by the local NOx concentration level.

We have also made some changes in the manuscript (Line 417-442).

**Revision:**

Line 417-442: Halogen species have been recognized as potent oxidizers that can boost photochemistry (Xia et al., 2022; Peng et al., 2021). A sensitivity test was performed by imposing BrO and IO into the base model to diagnose the impact of the halogen chemistry on the troposphere chemistry. The concentration of BrO and IO is set to ~5 ppt, which is a typical level in MBL site (Xia et al., 2022; Bloss et al., 2010; Whalley et al., 2010). The details of the mechanisms involved are listed in Tables S3 and S4. In this scenario (Fig. 4, green line). The daytime concentration of $HO_2$ radical decreased by 8.5% and 13.3% during the LAM and OCM periods, respectively, compared to the base model. However, there was no significant change in the concentration of OH radicals (<3%). Traditionally, it is believed that the inclusion of halogen chemistry leads to higher modeled OH concentrations and lower modeled $HO_2$ concentrations. Therefore, the lack of an increase in OH concentration with the introduction of the halogen mechanism at the YMK site calls for further investigation (Fig. S6). By modifying the NO concentration in different levels (Scenario 1: [NO]×150%, Scenario 2: base, Scenario 3: [NO]×20%, Scenario 4: [NO]×10%), the response of HOx radicals to the halogen mechanism varied under different NO levels. As the constrained NO

increased from 30 ppt to 500 ppt, the reduction in $HO_2$ radicals due to the Br and I mechanisms ranged between 10% and 20%. At elevated NOx levels, reactions between halogen radicals and NOx occurred, inhibiting the formation of OH radicals. In Scenario 1, the OH concentration even decreased by 3.5% when introducing the halogen mechanism. When NO concentration was constrained around 30 ppt (Scenario 4), similar to those obtained in RHaMBLe/CYPHEX campaigns, the modelled OH concentration increased by 14.4%, while the $HO_2$ concentration decreased by approximately 20.8% (Whalley et al., 2010; Bloss et al., 2010). Therefore, the sensitivity of OH radicals to the halogen mechanism in the YMK region is primarily limited by the local NOx concentration level.

*21. Fig. 4: This figure could be removed as figure 6 is more instructive.*
**Reply:**

Thanks for your suggestion. We have removed the previous Fig.4.

*22. Line 335 – 346: I'm not sure this case study adds anything to the paper as it stands and could be removed to make the paper more succinct.*
**Reply:**

Thanks for your suggestion. The case and the previous Fig.5 have been removed to make the paper more succinct.

*23. Line 358: I don't think the good agreement between modelled and measured HO2 should be used as an argument to exclude heterogeneous reactions in the model. If the inclusion of heterogeneous processes did reduce the modelled HO2 concentration, this could highlight missing HO2 sources in the model (or may indicate that some RO2 species present were detected as HO2) and so warrants investigation.*
**Reply:**

[Figure]

**Fig. 4.** Median diurnal profiles of the observed and modelled OH, HO₂, $k_{OH}$ during LAM and OCM episodes. The coloured shadows for OH and HO₂ radicals denote the 25 and 75% percentiles. The grey areas denote nighttime.

Thanks for your suggestion. We agree with the reviewer that the good agreement between modelled and measured HO₂ should not be used as an argument to exclude heterogeneous reactions in the model. We have added the removal path of HO₂ radicals by heterogeneous uptake. The inclusion of heterogeneous processes ($\gamma$ = 0.08) did reduce the modelled HO₂ concentration for ~10% during both LAM and OCM periods (Fig.4). This reduced agreement between observation and simulation emphasizes the presence of a missing HO₂ source in the base model.

**Revision:**

Line 277-289: The heterogeneous uptake of HO₂ is considered to play an important role in the MBL region (Whalley et al., 2010; Zou et al., 2022; Woodward-Massey et al., 2022). In order to assess the impact of HO₂ uptake on HOx radical chemistry, we incorporated HO₂ uptake reaction into the base model (Eq. (3) - (5)).

$$HO_2 + uptake \rightarrow products \tag{3}$$

$$k_{HO_2+uptake} = \frac{\gamma \times ASA \times v_{HO_2}}{k_{HO_2+NO}[NO]} \tag{4}$$

$$v_{HO_2} = \sqrt{\frac{8 \times R \times T}{0.033 \times \Pi}} \tag{5}$$

Here, ASA represents the aerosol surface area [$\mu m^2\ cm^{-3}$], which can be estimated as 20 times the $PM_{2.5}$ concentration [$\mu g/cm^3$]. $v_{HO_2}$ [$cm^{-1}$] can be calculated using Eq. (5), where T and R represent the temperature and gas constant, respectively. The heterogeneous uptake coefficien ($\gamma$) for $HO_2$ usually has high uncertainty, with typical values ranging from 0 to 1 (Song et al., 2021). In this study, we set $\gamma$ to 0.08 to evaluate the influence of $HO_2$ uptake on radical concentrations.

Line 443-448: Alough the modelled and measured $HO_2$ showed good agreement,the effect of $HO_2$ heterogeneous processes on the chemistry of HOx radicals is also worth exploring. The inclusion of heterogeneous processes ($\gamma$ = 0.08) did reduce the modelled $HO_2$ concentration for ~10% during both LAM and OCM periods (Fig. 4, yellow line). This reduced agreement between observation and simulation emphasizes the presence of a missing $HO_2$ source in the base model.

**24. Line 371: Given the model slightly overestimates HO2 and the calculated OH reactivity could be an underestimate of the total OH reactivity actually present, a missing OH source may be masked. A comment on these points should be provided.**

**Reply:**

Thanks for your suggestion. We added relevant comments on the missing OH sources.

**Revision:**

Line 404-405: Under enhanced photochemistry, the calculated OH reactivity could be an underestimation of the total OH reactivity, so a missing OH source may be masked.

**25. Line 394: D(OH) should be considered a lower limit as it uses calculated rather than measured kOH. This should be made clear.**

**Reply:**

Thanks for your suggestion. We emphasize this point in Line 467-470.

**Revision:**

Line 467-470: Because $k_{OH}$ was not measured during the observation experiment, the simulated value was used to analyze the removal rate. Therefore, D(OH) should be considered a lower limit as it uses calculated rather than measured $k_{OH}$ (Yang et al., 2022b).

**26. Line 421 – 423: Again, following on from my earlier comments, without a measurement of kOH, the absence of unknown OH recycling pathways can't be confirmed here.**

**Reply:**

Thanks for your suggestion. We modified the misleading description in Line 485-486.

**Revision:**

Line 485-486: When the simulated $k_{OH}$ was introduced into the experimental budgets, the difference between P(OH) and D(OH) was less than 2 ppb/h.

**27. Fig. 10, line 532 - 539: Some further details on how the model was run when it was used to predict ozone are needed. What model constraints were changed to variables other than ozone (presumably NO2 was also changed to a variable)? Why was the atmospheric lifetime changed from 18 hrs to 15 hrs and what was the rate of the first order loss term used? How did modelled OH, RO2 and HO2 change when the model was unconstrained to HONO?**

**Reply:**

Thanks for your suggestion. When using the model to predict ozone, both $O_3$ and NO were changed to variables. The 15 hrs in the manuscript was a clerical mistake, and the atmospheric lifetime used in base model or ozone prediction was 18 hrs. Assuming a boundary layer height of about 1 km, the rate of the first order loss term at 18 hrs is about 1.5 cm/s. The modelled OH, $HO_2$ and $RO_2$ change when the model was unconstrained to HONO were shown in Fig.S7. After evaluation, in LAM and OCM

sectors, concentration changes for OH were 46.9% and 43.2%, for $HO_2$ were 38.3% and 34.3%, for $RO_2$ were 43.7% and 39.0%, respectively.

[Figure]

**Fig. S7.** The modelled OH, $HO_2$ and $RO_2$ change when the model was unconstrained to HONO during LAM and OCM sectors, respectively.

**Revision:**

Line 255-258: The physical losses of species due to processes such as deposition, convection, and advection were approximately replaced by an 18 h atmospheric lifetime, corresponding to first order loss rate of ~1.5 cm/s (by assuming a boundary layer height of about 1 km).

Line 604-605: On the basis of the base scenario run, constraints of the observed ozone and NO concentrations were removed to predict ozone.

Line 588-591: The modelled OH, $HO_2$ and $RO_2$ change when the model was unconstrained to HONO were shown in Fig. S7. After evaluation, in LAM and OCM sectors, concentration changes for OH were 46.9% and 43.2%, for $HO_2$ were 38.3% and 34.3%, for $RO_2$ were 43.7% and 39.0%, respectively.

**28.** *Line 547 - 548: I'm not sure the findings from this study support this closing statement. Although the impact of HONO in this particular marine environment*

*is interesting, the elevated HONO concentrations are somewhat of an anomaly compared to the other marine environments. In regions where HONO concentrations are elevated, the sources of HONO would need to be identified to aid pollution mitigation policies.*

**Reply:**

Thank you for your suggestion. HONO measurements at the YMK site were conducted using a commercial Long-Path Absorption Photometer (LOPAP). The LOPAP method has been extensively tested for its suitability in detecting HONO in complex atmospheric conditions, as demonstrated in previous studies by (Yang et al., 2022a; Yang et al., 2021b; Wang et al., 2023). To ensure the accuracy of the measurements, zero air measurements were taken every 8 hours for a duration of 20 minutes to correct for instrument baseline fluctuations. This calibration procedure helps to minimize any potential biases and ensures the reliability of the HONO detection at the YMK site.

The high daytime HONO concentrations observed at the YMK site is a notable phenomenon. Given the location of the site, one possible contributor to the elevated HONO levels is emissions from cruise ships, as discussed in the study by (Sun et al., 2020). Additionally, other active tropospheric sources of HONO, such as heterogeneous reactions with $NO_2$ and photolysis of $p(NO_3^-)$ warrant consideration in the MBL area, as highlighted in the studies by (Zhu et al., 2022; Crilley et al., 2021).

*29. Line 564 – 567: These statements need to be supported by evidence or removed.*

**Reply:**

Thanks for your suggestion. These statements have been removed.

**Minor Comments**

1. *Line 84: '..heterogeneous iodine-organic chemistry' Could the authors provide the specific reactions they are referring to here.*

**Reply:**

    Thanks for your suggestion. (Huang et al., 2022) simulated the growth of particles with an aerosol model. The specific reactions considered in the model include Reactions as below:

$$IOP + H_2O \rightarrow HIO_3$$

$$HIO_3 + dimethylamine \rightarrow salt(low\ volatility/non-volatile)$$

$$glyoxal(or\ meso-erythritol) + HIO_3 \rightarrow organic\ acid(water\ soluble, low\ volatility)$$

$$organic\ acid + dimethylamine \rightarrow salt(non-volatile, stable\ salt)$$

2. *Line 155: change 'avoid' to 'reduce'*

**Reply:**

    Thanks for your suggestion. We have revised the manuscript as the reviewer's comment (Line 169).

3. *Line 183-184: I'm not sure about the terminology used here 'conventional pollutants', 'secondary pollutant precursors' and 'destruction products'. I suggest just listing all these species and not attempting to categorise them.*

**Reply:**

    Thanks for your suggestion. We have revised the manuscript as the reviewer's comment (Line 220-225).

4. *Line 183: 'carbonic oxide' to 'carbon monoxide'*

**Reply:**

    Thanks for your suggestion. We have revised the manuscript as the reviewer's comment (Line 223).

*5. Line 198: change 'radical related secondary pollution' to 'ozone'*

**Reply:**

Thanks for your suggestion. We have revised the manuscript as the reviewer's comment (Line 246).

*6. Line 199: remove 'conventional'*

**Reply:**

Thanks for your suggestion. We have revised the manuscript as the reviewer's comment (Line 248).

**Revision:**

The meteorological parameters, pollutants, and precursor concentrations mentioned in Section 2.2.2 were input into the model as boundary conditions.

*7. Line 249: 'grooved distribution' is strange terminology, I would delete.*

**Reply:**

Thanks for your suggestion. We have revised the manuscript as the reviewer's comment (Line 329-331).

*8. Line 252: 'extremely high..' 'significantly affect..' need to be more specific.*

**Reply:**

Thanks for your suggestion. We have revised the manuscript as the reviewer's comment (Line 333-334).

*9. Line 277 – 278: This needs rewording, as it is written, it could be interpreted as meaning the ozone and HONO concentrations were higher during the OCM period.*

**Reply:**

Thanks for your suggestion. We have deleted the misleading sentence.

*10. Line 280: '..changed greatly' I would be explicit, i.e. T increased..From figure 2, J(O1D) is very similar between the two sectors.*

**Reply:**

Thanks for your suggestion. We have revised the manuscript as the reviewer's comment (Line 323-324).

*11. Line 360 -368: Suggest referencing section 4.1 here*

**Reply:**

Thanks for your suggestion. We have revised the manuscript as the reviewer's comment (Line 390).

*12. Line 448: 'loss' to 'production'*

**Reply:**

Thanks for your suggestion. We have revised the manuscript as the reviewer's comment (Line 513).

*13. Fig 11: The YMK campaign is labelled as STORM-II in this fig. Change to YMK for consistency.*

**Reply:**

Thanks for your suggestion. We have changed the label to YMK for consistency.

[revised manuscript text omitted]

---

## Author Comment (AC2)

Dear Editor and Referee,

Thanks for your suggestions which significantly help us to improve the manuscript. Hereby, we submit our responses and the manuscript has been revised accordingly. If there are any further questions or comments, please let us know.

Best regards

Renzhi Hu on behalf of all co-authors

Key Lab. of Environmental Optics & Technology, Anhui Institute of Optics and Fine Mechanics, Chinese Academy of Sciences

230031 Hefei China

E-mail: rzhu@aiofm.ac.cn

**Major Comments**

*1. L 32-34: Average concentrations of OH and HO2 are provided for the LAM period in the abstract. For comparison, please also provide values for the OCM period.*

**Reply:**

Thanks for your suggestion. The average concentrations of OH and $HO_2$ are provided for the OCM period in the abstract (Line 29-33).

**Revision:**

Line 29-33: Under a typical ocean-atmosphere (OCM), reasonable measurement model agreement was achieved for both OH and $HO_2$ using a 0-D chemical box model incorporating the regional atmospheric chemistry mechanism version 2-Leuven isoprene mechanism (RACM2-LIM1), with daily averages of $4.5 \times 10^6$ $cm^{-3}$ and $4.9 \times 10^8$ $cm^{-3}$, respectively.

*2. L38-41: "After a sensitivity test, HONO-related chemistry elevated the ozone production rate by 33% and 39% during the LAM and OCM periods, respectively, while the nitric acid and sulfuric acid formation rates were 52% and 35% higher, respectively." – Please clarify the last part of this sentence. Are the nitric acid and sulfuric acid formation rate increases for the OCM or LAM period?*

**Reply:**

Thanks for your suggestion. The misleading sentence has been revised (line 41-44).

**Revision:**

Line 41-44: After a sensitivity test, HONO-related chemistry elevated the ozone production rate by 33% and 39% during the LAM and OCM periods, respectively. The nitric acid ($P(HNO_3)$) and sulfuric acid ($P(H_2SO_4)$) formation rates also increased simultaneously (~43% and ~48% for LAM and OCM sectors, respectively).

*3. L41-43: "The simulated daytime HONO and ozone concentrations were reduced*

*to a low level (~70 ppt and ~35 ppb) without the HONO constraint." – Are the reported concentrations for LAM, OCM or both periods together? For comparison, please also provide values simulated when HONO is constrained.*

**Reply:**

Thanks for your suggestion. The modelled concentrations (~70 ppt and ~35 ppb for HONO and $O_3$) are the diurnal average values during the whole observation. We have added the simulated values when HONO is constrained (Line 44-46).

**Revision:**

Line 44-46: Without the HONO constraint, simulated $O_3$ decreased from ~75 ppb to a global background (~35 ppb), and daytime HONO concentration were reduced to a low level (~70 ppt).

4. *L157-159: "A wavelength modulation for the background measurement that periodically switches from an on-resonant state to a non-resonant state has been widely used to obtain spectral zero." – Did the authors also used a chemical modulation approach as done now on most LIF-FAGE instruments to make sure that OH measurements are free from interferences? If so it should be discussed here. If not, the authors should comment on potential interferences on OH measurements.*

**Reply:**

Thanks for your suggestion. During the YMK campaign, we did not use a chemical modulation approach. We will discuss whether internal interference exists in AIOFM-LIF from the following aspects:

First of all, literature research shows that measurement interference is more related to the length of the inlet in the low-pressure cell (Griffith et al., 2016). In terms of system design, the AIOFM-LIF system uses a short-length inlet design to minimize this and other unknown disturbances (the distance from radical sampling to flourescence excitation is ~150 mm).

**Table.S1.** Comparison of key parameters related to ozonolysis reactions ($O_3$、alkenes、isoprene and NOx) between YMK and the intercomparison experiment. All the values are the diurnal average (10:00-15:00).

| Species | Intercomparison | YMK |
|---------|-----------------|-----|
| $O_3$ (ppb) | 71.02 | 74.58 |
| Alkenes (ppb) | 1.29 | 1.10 |
| Isoprene (ppb) | 0.67 | 0.64 |
| NOx (ppb) | 5.65 | 4.24 |

Additionally, potential interference may exist when the atmosphere contains abundant alkenes, ozone, and BVOCs, indicating that environmental conditions play leading roles in OH interferences (Mao et al., 2010; Fuchs et al., 2016; Novelli et al., 2014). An OH measurement comparison with a LIF instrument deployed an inlet pre-injector (PKU-LIF), was conducted in a real atmosphere in a previous study (Zhang et al., 2022b). The ozonolysis interference on the measurement consistency of both systems was excluded under high-VOCs conditions. We have compared the chemical conditions during the intercomparison experiment and the current environmental conditions. Overall, the key parameters related to ozonolysis reactions ($O_3$、alkenes、isoprene and NOx) in YMK were similar to those during the comparison experiment, which is not conducive to generating potential OH interference.

[Figure]

AIOFM-LIF have used a chemical modulation approach to examine the chemical

background of OH radicals in another field observation, Hefei, China. The specific description of the site is shown in (Ren et al., 2022). The environmental conditions during ozone pollution (2022.9.29-2022.10.3) are shown in the Figure above, with daytime peaks of ozone concentration above 75 ppb, accompanied by alkene species approaching ~10 ppb. The diurnal concentration of isoprene was also a high level (＞1 ppb). The chemical conditions are more favourable to induce OH interference than the YMK site (Table S1). However, the OH concentrations achieved by chemical modulation ($OH_{chem}$) and wavelength modulation ($OH_{wav}$) were in good agreement. No obvious chemical background was observed by deploying an inlet pre-injector. Therefore, it is not expected that OH measurement in the present study was affected by internal interference in the YMK site.

We added the detailed description in Line 177-187.

**Revision:**

Line 177-187: In terms of system design, the AIOFM-LIF system incorporates a short-length inlet design to minimize interferences from ozonolysis and other unknown factors (the distance from radical sampling to flourescence excitation is ~150 mm). An OH measurement comparison with an interference-free instrument, PKU-LIF, was conducted in a real atmosphere in a previous study (Zhang et al., 2022b). The ozonolysis interference on the measurement consistency of both systems was excluded under high-VOCs condition. Overall, the key parameters related to ozonolysis reactions ($O_3$、alkenes、isoprene and NOx) in YMK was similar to that during the intercomparison experiment, implies that the chemical conditions do not favor the generation of potential interference to OH measurement (Table S1).

5. *"The ozone photolysis interference was subtracted according to laboratory experiments." – What was the contribution of this interference to the total measured OH signal (interference + ambient)?*

**Reply:**

Thanks for your suggestion. We have added the Fig. S2, and the detailed

description was in Line 173-177.

**Revision:**

Line 173-177: Due to the synchronous reaction at 308nm, wavelength modulation is not applicable to ozone photolysis interference. Through laboratory experiments, at 20 mW laser energy, every 1% water vapor concentration and 50 ppb ozone concentration can generate a $2.5 \times 10^5$ cm$^{-3}$ OH concentration. The results in this paper have subtracted the ozone photolysis interference (Fig. S2).

[Figure]

**Fig. S2.** Mean diurnal profiles of measured [OH] before (red line) and after (blue line) deducting the $O_3$ interference. The coloured shadows denote the 25 and 75% percentiles. The grey areas denote nighttime.

6. *"The ozonolysis interference on the measurement consistency of both systems was excluded under high-NOx and high-NMHC conditions, confirming the general applicability under complex atmospheric pollution." – What do the authors mean by "ozonolysis interference"? What type of interference is it? The authors indicate that they could rule out interferences under high-NOx and high-NMHC conditions from a comparison with an interference free instrument. What about low-NOx conditions as encountered in the MBL? Why do the authors consider PKU-LIF to be free of interferences?*

**Reply:**

The term "ozonolysis interference" refers to a potential interference that can affect LIF-FAGE measurements of ambient OH. It is important to note that this type of interference is internally generated within the detection cell of the measurement system.

This interference arises from the ozonolysis of biogenic alkenes, as described in previous studies by Mao et al. (2012) and Rickly and Stevens (2018). The occurrence of ozonolysis interference depends on the system design and environmental conditions, particularly when the atmosphere contains significant amounts of ozone, alkenes, and BVOCs (Mao et al., 2010; Fuchs et al., 2016; Novelli et al., 2014).

The PKU-LIF system has been utilized for measuring HOx concentrations in various campaigns, and a chemical modulation approach has been employed since 2014 to quantify potential interferences (Ma et al., 2022; Yang et al., 2021; Tan et al., 2019; Tan et al., 2018a; Tan et al., 2017a). These prior studies have demonstrated that no significant internal interference existed in the PKU-LIF system, indicating that its accuracy has already been established.

In the previous comprehensive comparison experiment, AIOFM-LIF and PKU-LIF were compared under multiple conditions, including high NOx, high VOCs, low NOx, and high BVOCs. The results showed that changes in environmental conditions did not affect the measurement consistency between the two systems. Considering the key parameters related to ozonolysis reactions ($O_3$、alkenes、isoprene and NOx) in YMK was similar to that during the intercomparison experiment, we determined that the chemical conditions do not favor the generation of potential interference to OH measurements (Table S1).

7. *L164-167: "For HO2 measurement, the NO concentration corresponding to a conversion efficiency of ~15% was selected to avoid RO2→HO2 interference (especially from RO2 radicals derived from long chain alkanes (C ≥ 3), alkenes, and aromatic hydrocarbons." – The authors optimized operating conditions to minimize this interference. However, to this reviewer's knowledge, it is not possible to completely eliminate this interference. The authors should comment on the level of interference that is still expected from the most abundant RO2 radicals at the measurement site. If a significant interference is expected, the authors should report this measurement as HO2\* and should compare it to*

*modelled HO2\* values instead of HO2.*

**Reply:**

Thank you for your response. We acknowledge and agree with the reviewer's perspective that it is challenging to completely eliminate the interference caused by $RO_2$ conversion. In the previous work, we have calculated the conversion efficiency of alkene-derived $RO_2$ to OH under different NO concentration (Wang et al., 2021). In this observation, ethene accounted for about 70% of the total ethene concentration (Table S5). Therefore, we choose ethene and isoprene to investigate the percentage interference from an alkene-derived $RO_2$. When NO was at $1.6 \times 10^{12}$ cm$^{-3}$, the conversion efficiency of $HO_2$ was ~15%, and the percentage interference from ethene and isoprene-derived $RO_2$ was 3.83% and 1.75%, respectively (Wang et al., 2021). We added the detailed description in Line 187-195.

**Revision:**

Line 187-195: For $HO_2$ measurement, the NO gas was mixed with 2% in $N_2$ to achieve $HO_2$-to-OH conversion. NO was passed through a ferrous sulfate filter to remove impurities ($NO_2$, HONO, and so on) before being injected into the detection cell. The NO concentration (~$1.6 \times 10^{12}$ cm$^{-3}$) corresponding to a conversion efficiency of ~15% was selected to avoid $RO_2 \rightarrow HO_2$ interference (especially from $RO_2$ radicals derived from long-chain alkanes (C ≥ 3), alkenes, and aromatic hydrocarbons). Previous study denoted that the percentage interference from alkene-derived $RO_2$ under these operating conditions was no more than 5% (Wang et al., 2021).

8. *L175: "measurement errors were 13% and 17%" – Please clarify in the text how these values were assessed? If these values are derived from uncertainties associated to the generated radical concentrations it should read "measurement accuracy"*

**Reply:**

Thanks for your suggestion. We acknowledge and agree with the reviewer's perspective that the "measurement errors" should be changed as "measurement

accuracy". We determine the value by considering the system uncertainty and calibration uncertainty, and the measurement accuracy for OH and HO$_2$ were 13% and 17%, respectively. We added the detailed description in Line 213-216.

**Revision:**

Line 213-216: Considering the system uncertainty and calibration uncertainty, the detection limits of the OH and HO$_2$ radicals were $3.3 \times 10^5$ cm$^{-3}$ and $1.1 \times 10^6$ cm$^{-3}$ (60 s, $1\sigma$), respectively. At a typical laser power of 15 mW, the measurement accuracy for OH and HO$_2$ measurement was 13% and 17%, respectively.

*9. L185-191: The authors should provide more details on the measured VOCs in the supplementary material. What were the most abundant species in each category (alkanes, alkenes, aromatics, OVOCs)? What was the campaign averaged concentration of each category? Etc.*

**Reply:**

Thanks for your suggestion. The detailed information for VOCs species during the YMK campaign has been added in the Supplement (Table. S5). We added the detailed description in Line 338-339.

**Revision:**

Line 338-339: The detailed information for VOCs species during the YMK campaign has been added in the Table S5.

**Table. S5.** The detailed information table for VOCs species during the YMK campaign. The mean concentration, standard deviation (SD), minimum value (Min), maximum value (Max), and percentage contribution in the species for the top-five ranked species in alkanes, alkenes, aromatic and OVOCs are listed. All the values are the daily average (0:00-24:00).

| Species | Mean (ppb) | Sd (ppb) | Min (ppb) | Max (ppb) | Proportion (%) |
|---|---|---|---|---|---|
| Alkane | | | | | |
| ethane | 1.72 | 0.564 | 0.24 | 5.621 | 29.2 |
| propane | 1.246 | 0.524 | 0.136 | 5.438 | 21.15 |
| n-butane | 0.646 | 0.395 | 0.054 | 2.424 | 10.97 |
| i-butane | 0.561 | 0.471 | 0.029 | 3.372 | 9.52 |
| n-hexane | 0.41 | 0.307 | 0.033 | 3.026 | 6.96 |
| Alkene | | | | | |
| ethene | 0.592 | 0.656 | 0.034 | 5.48 | 69.08 |
| propene | 0.123 | 0.127 | 0.017 | 1.187 | 14.35 |

| | | | | | |
|---|---|---|---|---|---|
| 1-butene | 0.046 | 0.014 | 0.012 | 0.107 | 5.37 |
| trans-2-butene | 0.028 | 0.006 | 0.006 | 0.05 | 3.27 |
| cis-2-butene | 0.026 | 0.006 | 0.007 | 0.045 | 3.03 |
| Aromatic | | | | | |
| toluene | 0.523 | 0.361 | 0.035 | 2.82 | 38.34 |
| benzene | 0.286 | 0.112 | 0.032 | 0.742 | 20.97 |
| m-xylene | 0.123 | 0.237 | 0.015 | 3.579 | 9.02 |
| ethyl benzene | 0.107 | 0.134 | 0.017 | 2.052 | 7.84 |
| o-xylene | 0.103 | 0.214 | 0.015 | 3.294 | 7.55 |
| OVOC | | | | | |
| acetone | 3.297 | 0.835 | 0.412 | 5.978 | 52.47 |
| acetaldehyde | 1.742 | 0.635 | 0.276 | 5.805 | 27.73 |
| methyl ethyl ketone | 0.496 | 0.15 | 0.051 | 1.118 | 7.89 |
| methyl t-butyl ether | 0.213 | 0.208 | 0.018 | 1.512 | 3.39 |
| propionaldehyde | 0.178 | 0.081 | 0.028 | 0.572 | 2.83 |

*10. L191: "All of the instruments were located close to the roof of the fourth floor" – It was not indicated in the text before that there is a building at the measurement site. Please provide some details in the site description section.*

**Reply:**

Thanks for your suggestion. The site is a part of Shenzhen ecological monitoring Center station, approximately 35 m above sea level, and the sea is approximately 150 m to the east. All of the instruments were located close to the roof of the monitoring building. We added the detailed description in Line 117-119&240-242.

**Revision:**

Line 118-120: The site is a part of Shenzhen Ecological Monitoring Center station, approximately 35 m above sea level, and the sea is approximately 150 m to the east.

Line 240-242: All of the instruments were located close to the roof of the monitoring building, nearly 12 m above the ground to ensure that all of the pollutants were located in a homogeneous air mass.

*11. L202-204: "The overall average during the observations was substituted for large areas of missing data due to instrument maintenance or failure." – How long were these time periods? They should be highlighted in Figure 3. It is interesting*

*to note that while using campaign average data when ancillary measurements are missing could lead to improper model constraint, it does not appear to have a significant impact on the model-measurement agreement.*

**Reply:**

Thanks for your suggestion. Considering the instrument failure of GC-MS in 10.24-10.26, we use the overall average data to fill the missing VOCs data. We have identified the time interval of the missing data in Fig. S3.

**Revision:**

[Figure]

**Fig. S3.** Time series of observed meteorological and chemical parameters at YMK from18 October to October 28, 2019. The GC-MS instrument failed between 24 and 26 October, and the missing VOCs data were replaced by the average value during the observation period. Only isoprene was considered in the BVOCs contribution.

*12. L210: "the simulation accuracy of the model for the OH and HO2 radicals was 50%" – Please specify if this is 1 or 2 σ*

**Reply:**

Thanks for your suggestion. The simulation accuracy of the model for the OH and HO$_2$ radicals was 50%, 1σ.

*13. L211-217: The bromine chemistry is included in the chemical mechanism to test the HOx sensitivity. What about the iodine chemistry? Is there a specific reason why it was not included in the mechanism as well?*

**Reply:**

In response to the reviewer's suggestion, Iodine-related mechanisms are also considered in the latest version of the manuscript. In order to better explore the effect of Br and I chemistry on HOx radicals, we chose BrO/IO as the initiation point of halogen chemistry. The concentration of BrO and IO is set to ~5 ppt, which is a typical level in MBL site (Xia et al., 2022; Bloss et al., 2010; Whalley et al., 2010).

[Figure]

**Fig. 4.** Median diurnal profiles of the observed and modelled OH, $HO_2$, $k_{OH}$ during LAM and OCM episodes. The coloured shadows for OH and $HO_2$ radicals denote the 25 and 75% percentiles. The grey areas denote nighttime.

In this scenario (Fig. 4, green line). The daytime concentration of $HO_2$ radical decreased by 8.5% and 13.3% during the LAM and OCM periods, respectively, compared to the base model. However, there was no significant change in the concentration of OH radicals (<3%). We added the detailed description in Line 270-276&417-426.

**Revision:**

Line 270-276: Considering the environmental characteristics of the MBL, the gas-phase mechanisms for bromine (Br) and iodine (I) were introduced into the base model to diagnose the impacts of the reactive bromine chemistry. The details of the mechanisms involved are listed in Tables S3 and S4. The halogen species were not available in the YMK site, so the typical levels of BrO and IO concentration in MBL site was used as a reference value (average daytime concentration of ~5 ppt) (Xia et al., 2022; Bloss et al., 2010; Whalley et al., 2010).

Line 417-426: Halogen species have been recognized as potent oxidizers that can boost photochemistry (Xia et al., 2022; Peng et al., 2021). A sensitivity test was performed by imposing BrO and IO into the base model to diagnose the impact of the halogen chemistry on the troposphere chemistry. The concentration of BrO and IO is set to ~5 ppt, which is a typical level in MBL site (Xia et al., 2022; Bloss et al., 2010; Whalley et al., 2010). The details of the mechanisms involved are listed in Tables S3 and S4. In this scenario (Fig. 4, green line). The daytime concentration of $HO_2$ radical decreased by 8.5% and 13.3% during the LAM and OCM periods, respectively, compared to the base model. However, there was no significant change in the concentration of OH radicals (<3%).

*14. L301-302 & Fig. 3: How does the modelled kOH compare to that calculated from the model constrains? How much OH reactivity does the model generate from unconstrained OVOCs? Since VOCs are constrained as lumped groups in RACM, OH reactivity from unmeasured OVOCs may be underestimated. Could the authors comment on this?*

**Reply:**

In response to the reviewer's suggestion, we have adopted a classification for the $k_{OVOCs}$, separating them into $k_{OVOCs(Obs)}$ and $k_{OVOCs(Model)}$. Specifically, $k_{OVOCs(Obs)}$ includes the observed species such as formaldehyde (HCHO), acetaldehyde (ACD), higher aldehydes (ALD), acetone (ACT), ketones (KET), and oxidation products of isoprene (MACR and MVK). The model-generated intermediates, such as glyoxal,

methylglyoxal, methylethyl ketone, and methanol, are categorized as $k_{OVOCs(Model)}$. Approximately 50% of the total $k_{OVOCs}$ are represented by unconstrained species ($k_{OVOCs(Model)}$), which contribute a daily $k_{OH}$ of 1.39 s$^{-1}$. It should be noted that the OH reactivity of unmeasured VOCs may be underestimated due to the lumped groups in RACM. We have updated Fig.4 to include this classification of $k_{OVOCs}$. We added the detailed description in Line 362-375.

**Revision:**

Line 362-375: The $k_{OVOCs}$ was separated into $k_{OVOCs(Obs)}$ and $k_{OVOCs(Model)}$ (Fig. 3(c)). Specifically, $k_{OVOCs(Obs)}$ includes the observed species such as formaldehyde (HCHO), acetaldehyde (ACD), higher aldehydes (ALD), acetone (ACT), ketones (KET), and oxidation products of isoprene (MACR and MVK). The model-generated intermediates, such as glyoxal, methylglyoxal, methylethyl ketone, and methanol, are categorized as $k_{OVOCs(Model)}$. Approximately 50% of the total $k_{OVOCs}$ are represented by unconstrained species ($k_{OVOCs(Model)}$), which contribute a daily $k_{OH}$ of 1.39 s$^{-1}$. Overall, the observed OH and HO$_2$ concentrations were both well reproduced by the base model incorporating the RACM2-LIM1 mechanism. The observed OH was underestimated only on the first days, and a slight model overestimation happened on October 23&24. PSS calculation showed good agreement with the base model, providing evidence of the balance of radical internal consistency in the daytime. It should be noted that the OH reactivity of unmeasured VOCs may be underestimated due to the lumped groups in RACM2 mechanism.

*15. L311-313: "The base model slightly overestimated the OH radical, suggesting that a radical removal pathway was missing." – The authors should this statement. The measurement/model agreement is well within uncertainty. In addition, this is only observed on the first 2 days and a model underestimation is observed on 10/23 & 10/24.*

**Reply:**

Thanks for your suggestion. We have removed the statement (Line 369-372).

**Revision:**

Line 369-372: Overall, the observed OH and HO$_2$ concentrations were both well reproduced by the base model incorporating the RACM2-LIM1 mechanism. The observed OH was underestimated only on the first days, and a slight model overestimation happened on October 23&24.

*16. L314-327: Model sensitivity to halogen chemistry - What was the range of BrO concentrations simulated by the model? Is it comparable to BrO concentrations measured in the MBL? As mentioned in a previous comment, iodine chemistry was not added in the model. Why? Could the authors comment on the potential impact of this chemistry?*

**Reply:**

Thanks for your suggestion. In the previous manuscript, when the model was run with Br$_2$ chemistry, the diurnal concentration of BrO was depicted in the following Figure. During the observation period, BrO concentration exhibited a clear diurnal variation with peak concentrations at 0.68 ppt. This value is consistent with the simulated results observed by HZ (~0.5 ppt) but lower than those obtained at CHABLIS (~5.0 ppt) (Bloss et al., 2010; Xia et al., 2022).

[Figure]

In response to the reviewer's suggestion, Iodine-related mechanisms are also considered in the latest version of the manuscript. In order to better explore the effect of Br and I chemistry on HOx radicals, we chose BrO/IO as the initiation point of halogen chemistry. The concentration of BrO and IO is set to ~5 ppt, which is a typical level in MBL site (Xia et al., 2022; Bloss et al., 2010; Whalley et al., 2010).

[Figure]

**Fig. 4.** Median diurnal profiles of the observed and modelled OH, HO₂, $k_{OH}$ during LAM and OCM episodes. The coloured shadows for OH and HO₂ radicals denote the 25 and 75% percentiles. The grey areas denote nighttime.

In this scenario (Fig. 4, green line). The daytime concentration of HO₂ radical decreased by 8.5% and 13.3% during the LAM and OCM periods, respectively, compared to the base model. However, there was no significant change in the concentration of OH radicals (<3%). We added the detailed description in Line 417-426.

**Revision:**

Line 417-426: Halogen species have been recognized as potent oxidizers that can boost photochemistry (Xia et al., 2022; Peng et al., 2021). A sensitivity test was performed by imposing BrO and IO into the base model to diagnose the impact of the halogen chemistry on the troposphere chemistry. The concentration of BrO and IO is set to ~5 ppt, which is a typical level in MBL site (Xia et al., 2022; Bloss et al., 2010; Whalley et al., 2010). The details of the mechanisms involved are listed in Tables S3 and S4. In this scenario (Fig. 4, green line). The daytime concentration of HO₂ radical decreased by 8.5% and 13.3% during the LAM and OCM periods, respectively, compared to the base model. However, there was no significant change in the concentration of OH radicals (<3%).

*17. L331-346 & Fig. 5: This reviewer does not see the added value of this section and thinks that it moves the reader's focus away from the main results. It is suggested to remove it.*

**Reply:**

Thanks for your suggestion. The case and the previous Fig.5 have been removed to make the paper more succinct.

*18. Eq. 3: The second term on the right-hand side should include the organic nitrate yield from RO2+NO. The authors may need to recalculate P(Ox) values displayed in Fig. 9 if the organic nitrate yield was not considered.*

**Reply:**

Thanks for your suggestion. When calculating P(Ox) in the previous Fig.9, the contribution from the formation of organic nitrates has been subtracted. This portion of the side reaction process is denoted in the previous Eq.3. We added the detailed description in Line 527-528.

**Revision:**

Line 522, Eq.8:

$$F(O_x) = k_{HO_2+NO}[NO][HO_2] + \sum_i (1-\alpha_i) k_{RO_2^i+NO}[NO]RO_2^i \qquad (8)$$

Line 527-528: $\alpha_i$ represents the side generation ratio of organic nitrate, which also affects the quantum yield of $NO_2$ (Tan et al., 2018b).

*19. L529-542: Please provide details on the time dependent box model in the supplementary material.*

**Reply:**

Thanks for your suggestion. The details on the time dependent box model have been added to the supplementary material (Text S1).

**Revision:**

**S1 Brief overview of the ozone-prediction mode in box model**

A 0-D chemical box model incorporating a condensed mechanism, the regional atmospheric chemistry mechanism version 2-Leuven isoprene mechanism (RACM2-LIM1), was used to predict ozone concentration (Stockwell et al., 1997; Griffith et al., 2013; Tan et al., 2017b). In the ozone-estimation mode, the meteorological parameters, pollutants, and precursor concentrations mentioned in Section 2.2.2 were input into the model as boundary conditions, and the temporal resolution for all of the constraints was unified to 15 min. Three days of data were entered in advance as the spin-up period. The hydrogen ($H_2$) and methane ($CH_4$) concentrations were set to fixed values of 550 ppb and 1900 ppb, respectively. The physical losses of species due to processes such as deposition, convection, and advection were approximately replaced by an 18 h atmospheric lifetime, corresponding to a first-order loss rate of ~1.5 cm/s. Constraints of the observed ozone and NO concentrations were removed on the basis of the base scenario. According to the measurement accuracy, the simulation accuracy of the model for the OH and $HO_2$ radicals was 50% (Zhang et al., 2022a). To specifically quantify the contribution of HONO-induced ozone generation, a sensitivity test was conducted without constraints on HONO (i.e., w.o HONO). Only the homogeneous reaction (OH + NO) participated in the formation of HONO in the default mode without HONO input.

**Minor Comments**

*1. L183: "carbonic oxide" should read "carbon monoxide"*

**Reply:**

Thanks for your suggestion. We have revised the manuscript as the reviewer's comment (Line 223).

*2. L271: "Serval observation campaign" should read "Several observation campaigns"*

**Reply:**

Thanks for your suggestion. We have revised the manuscript as the reviewer's comment.

*3. L299: Since a range of concentrations is given for both OH and HO2, "The average daily maximum" should read "The daily maximum". Other instances in the text.*

**Reply:**

Thanks for your suggestion. We have revised the manuscript as the reviewer's comment in Line 26&339&356&626.

*4. L332: Please define ROx*

**Reply:**

Thanks for your suggestion. We have revised the manuscript as the reviewer's comment in Line 400.

*5. 2: Please define the different parameters*

**Reply:**

Thanks for your suggestion. We have revised the manuscript as the reviewer's comment in Line 463-464.

**Revision:**

Line 463-464: Here, $\varphi_{OH}$ and $\varphi_{OH}^{i}$ represent the OH yields in the $O_3$ photolysis and alkene ozonolysis processes, respectively.

6. *L447-448: "As the only known gas-phase source, OH + NO accounted for a negligible proportion of the HONO loss." Should read "As the only known gas-phase source, OH + NO accounted for a negligible proportion of the HONO production rate."*

**Reply:**

Thanks for your suggestion. We have revised the manuscript as the reviewer's comment in Line 512-513.

7. *L455: "Peroxyl radical" should read "Peroxy radical". Other instances in the text.*

**Reply:**

Thanks for your suggestion. We have revised the manuscript as the reviewer's comment.

8. *L573: "peroxynitrite" should read "peroxynitrate"*

**Reply:**

Thanks for your suggestion. We have revised the manuscript as the reviewer's comment in Line 504&643.

9. *Fig S2: Please indicate the color code for back-trajectories*

**Reply:**

Thanks for your suggestion. We have revised the manuscript as the reviewer's comment.

**Revision:**

Fig. S1.: The 24-h backward trajectories calculated at an arrival time of 12:00 (local time) at 100 m (red line), 500 (blue line), 1000 m (green line) above ground level at YMK in special days;

**References**

Bloss, W. J., Camredon, M., Lee, J. D., Heard, D. E., Plane, J. M. C., Saiz-Lopez, A., Bauguitte, S. J. B., Salmon, R. A., and Jones, A. E.: Coupling of HOx, NOx and halogen chemistry in the antarctic boundary layer, Atmos Chem Phys, 10, 10187-10209, 10.5194/acp-10-10187-2010, 2010.

Fuchs, H., Tan, Z., Hofzumahaus, A., Broch, S., Dorn, H.-P., Holland, F., Kuenstler, C., Gomm, S., Rohrer, F., Schrade, S., Tillmann, R., and Wahner, A.: Investigation of potential interferences in the detection of atmospheric ROx radicals by laser-induced fluorescence under dark conditions, Atmos Meas Tech, 9, 1431-1447, 10.5194/amt-9-1431-2016, 2016.

Griffith, S. M., Hansen, R. F., Dusanter, S., Stevens, P. S., Alaghmand, M., Bertman, S. B., Carroll, M. A., Erickson, M., Galloway, M., Grossberg, N., Hottle, J., Hou, J., Jobson, B. T., Kammrath, A., Keutsch, F. N., Lefer, B. L., Mielke, L. H., O'Brien, A., Shepson, P. B., Thurlow, M., Wallace, W., Zhang, N., and Zhou, X. L.: OH and HO2 radical chemistry during PROPHET 2008 and CABINEX 2009-Part 1: Measurements and model comparison, Atmos Chem Phys, 13, 5403-5423, 10.5194/acp-13-5403-2013, 2013.

Griffith, S. M., Hansen, R. F., Dusanter, S., Michoud, V., Gilman, J. B., Kuster, W. C., Veres, P. R., Graus, M., de Gouw, J. A., Roberts, J., Young, C., Washenfelder, R., Brown, S. S., Thalman, R., Waxman, E., Volkamer, R., Tsai, C., Stutz, J., Flynn, J. H., Grossberg, N., Lefer, B., Alvarez, S. L., Rappenglueck, B., Mielke, L. H., Osthoff, H. D., and Stevens, P. S.: Measurements of hydroxyl and hydroperoxy radicals during CalNex-LA: Model comparisons and radical budgets, J Geophys Res-Atmos, 121, 4211-4232, 10.1002/2015jd024358, 2016.

Ma, X., Tan, Z., Lu, K., Yang, X., Chen, X., Wang, H., Chen, S., Fang, X., Li, S., Li, X., Liu, J., Liu, Y., Lou, S., Qiu, W., Wang, H., Zeng, L., and Zhang, Y.: OH and HO2 radical chemistry at a suburban site during the EXPLORE-YRD campaign in 2018, Atmos Chem Phys, 22, 7005-7028, 10.5194/acp-22-7005-2022, 2022.

Mao, J., Ren, X., Chen, S., Brune, W. H., Chen, Z., Martinez, M., Harder, H., Lefer, B., Rappenglück, B., Flynn, J., and Leuchner, M.: Atmospheric oxidation capacity in the summer of Houston 2006: Comparison with summer measurements in other metropolitan studies, Atmos Environ, 44, 4107-4115, 10.1016/j.atmosenv.2009.01.013, 2010.

Novelli, A., Hens, K., Ernest, C. T., Kubistin, D., Regelin, E., Elste, T., Plass-Duelmer, C., Martinez, M., Lelieveld, J., and Harder, H.: Characterisation of an inlet pre-injector laser-induced fluorescence instrument for the measurement of atmospheric hydroxyl radicals, Atmos Meas Tech, 7, 3413-3430, 10.5194/amt-7-3413-2014, 2014.

Peng, X., Wang, W. H., Xia, M., Chen, H., Ravishankara, A. R., Li, Q. Y., Saiz-Lopez, A., Liu, P. F., Zhang, F., Zhang, C. L., Xue, L. K., Wang, X. F., George, C., Wang, J. H., Mu, Y. J., Chen, J. M., and Wang, T.: An unexpected large continental source of reactive bromine and chlorine with significant impact on wintertime air quality, Natl. Sci. Rev., 8, 10.1093/nsr/nwaa304, 2021.

Ren, B., Xie, P. H., Xu, J., Li, A., Qin, M., Hu, R. Z., Zhang, T. S., Fan, G. Q., Tian, X., Zhu, W., Hu, Z. K., Huang, Y. Y., Li, X. M., Meng, F. H., Zhang, G. X., Tong, J. Z., Ren, H. M., Zheng, J. Y., Zhang, Z. D., and Lv, Y. S.: Vertical characteristics of NO2 and HCHO, and the ozone formation regimes in Hefei, China, Sci Total Environ, 823, 10.1016/j.scitotenv.2022.153425, 2022.

Stockwell, W. R., Kirchner, F., Kuhn, M., and Seefeld, S.: A new mechanism for regional atmospheric chemistry modeling, J Geophys Res-Atmos, 102, 25847-25879, 10.1029/97jd00849, 1997.

Tan, Z., Fuchs, H., Lu, K., Hofzumahaus, A., Bohn, B., Broch, S., Dong, H., Gomm, S., Haeseler, R., He, L., Holland, F., Li, X., Liu, Y., Lu, S., Rohrer, F., Shao, M., Wang, B., Wang, M., Wu, Y., Zeng, L., Zhang, Y., Wahner, A., and Zhang, Y.: Radical chemistry at a rural site (Wangdu) in the North China Plain: observation and model calculations of OH, HO2 and RO2 radicals, Atmos Chem Phys, 17, 663-690, 10.5194/acp-17-663-2017, 2017a.

Tan, Z., Rohrer, F., Lu, K., Ma, X., Bohn, B., Broch, S., Dong, H., Fuchs, H., Gkatzelis, G. I., Hofzumahaus, A., Holland, F., Li, X., Liu, Y., Liu, Y., Novelli, A., Shao, M., Wang, H., Wu, Y., Zeng, L., Hu, M., Kiendler-Scharr, A., Wahner, A., and Zhang, Y.: Wintertime photochemistry in Beijing: observations of ROx radical concentrations in the North China Plain during the BEST-ONE campaign, Atmos Chem Phys, 18, 12391-12411, 10.5194/acp-18-12391-2018, 2018a.

Tan, Z. F., Lu, K. D., Dong, H. B., Hu, M., Li, X., Liu, Y. H., Lu, S. H., Shao, M., Su, R., Wang, H. C., Wu, Y. S., Wahner, A., and Zhang, Y. H.: Explicit diagnosis of the local ozone production rate and the ozone-NOx-VOC sensitivities, Sci. Bull., 63, 1067-1076, 10.1016/j.scib.2018.07.001, 2018b.

Tan, Z. F., Lu, K. D., Hofzumahaus, A., Fuchs, H., Bohn, B., Holland, F., Liu, Y. H., Rohrer, F., Shao, M., Sun, K., Wu, Y. S., Zeng, L. M., Zhang, Y. S., Zou, Q., Kiendler-Scharr, A., Wahner, A., and Zhang, Y. H.: Experimental budgets of OH, HO2, and RO2 radicals and implications for ozone formation in the Pearl River Delta in China 2014, Atmos Chem Phys, 19, 7129-7150, 10.5194/acp-19-7129-2019, 2019.

Tan, Z. F., Fuchs, H., Lu, K. D., Hofzumahaus, A., Bohn, B., Broch, S., Dong, H. B., Gomm, S., Haseler, R., He, L. Y., Holland, F., Li, X., Liu, Y., Lu, S. H., Rohrer, F., Shao, M., Wang, B. L., Wang, M., Wu, Y. S., Zeng, L. M., Zhang, Y. S., Wahner, A., and Zhang, Y. H.: Radical chemistry at a rural site (Wangdu) in the North China Plain: observation and model calculations of OH, HO2 and RO2 radicals, Atmos Chem Phys, 17, 663-690, 10.5194/acp-17-663-2017, 2017b.

Wang, Y., Hu, R., Xie, P., Chen, H., Wang, F., Liu, X., Liu, J., and Liu, W.: Measurement of tropospheric HO2 radical using fluorescence assay by gas expansion with low interferences, J Environ Sci (China), 99, 40-50, 10.1016/j.jes.2020.06.010, 2021.

Whalley, L. K., Furneaux, K. L., Goddard, A., Lee, J. D., Mahajan, A., Oetjen, H., Read, K. A., Kaaden, N., Carpenter, L. J., Lewis, A. C., Plane, J. M. C., Saltzman, E. S., Wiedensohler, A., and Heard, D. E.: The chemistry of OH and HO2 radicals in the boundary layer over the tropical Atlantic Ocean, Atmos Chem Phys, 10, 1555-1576, 2010.

Xia, M., Wang, T., Wang, Z., Chen, Y., Peng, X., Huo, Y., Wang, W., Yuan, Q., Jiang, Y., Guo, H., Lau, C., Leung, K., Yu, A., and Lee, S.: Pollution-Derived Br2 Boosts Oxidation Power of the Coastal Atmosphere, Environ Sci Technol, 10.1021/acs.est.2c02434, 2022.

Yang, X., Lu, K., Ma, X., Liu, Y., Wang, H., Hu, R., Li, X., Lou, S., Chen, S., Dong, H., Wang, F., Wang, Y., Zhang, G., Li, S., Yang, S., Yang, Y., Kuang, C., Tan, Z., Chen, X., Qiu, P., Zeng, L., Xie, P., and Zhang, Y.: Observations and modeling of OH and HO2 radicals in Chengdu, China in summer 2019, The Science of the total environment, 772, 144829-144829, 10.1016/j.scitotenv.2020.144829, 2021.

Zhang, G., Hu, R., Xie, P., Lou, S., Wang, F., Wang, Y., Qin, M., Li, X., Liu, X., Wang, Y., and Liu, W.: Observation and simulation of HOx radicals in an urban area in Shanghai, China, Sci Total Environ, 810, 152275, 10.1016/j.scitotenv.2021.152275, 2022a.

Zhang, G., Hu, R., Xie, P., Lu, K., Lou, S., Liu, X., Li, X., Wang, F., Wang, Y., Yang, X., Cai, H., Wang, Y., and Liu, W.: Intercomparison of OH radical measurement in a complex atmosphere in Chengdu, China, Sci Total Environ, 155924, 10.1016/j.scitotenv.2022.155924, 2022b.

---

## Author Response (AR1)

Dear Editor and Referee,

Thanks for your suggestions which significantly help us to improve the manuscript. Hereby, we submit our responses and the manuscript has been revised accordingly. If there are any further questions or comments, please let us know.

**Best regards**

Renzhi Hu on behalf of all co-authors Key Lab. of Environmental Optics & Technology, Anhui Institute of Optics and Fine Mechanics, Chinese Academy of Sciences 230031 Hefei China E-mail: rzhu@aiofm.ac.cn

**Reviewer #1 (Major Comments)**

1. In section 2, the authors should describe how the OCM and LAM sectors were assigned – by wind direction or trajectory analysis? Some of this description is provided in lines 261 – 270 and I suggest this is moved to section 2.

**Reply:**

Thanks for your suggestion. The OCM and LAM sectors were assigned by trajectory analysis. We moved the detailed description to Section 2 (Line 132-140).

**Revision:**

Line 132-140: Using the hybrid single-particle Lagrangian integrated trajectory (HYSPLIT) model, the 24-h backward trajectories on special days were obtained. In Fig. S1, the red, blue, and green trajectories represent the results at altitudes of 100, 500, and 1000 m above ground level, respectively. Two typical transportation pathways dominated the air parcels. One originated from the northern megacities in the Pearl River Delta (defined as the land mass, LAM), especially on October 18, 19, and 27. In contrast, a clean air mass from the east or northeast was mainly transported to the observation site from the ocean (defined as the ocean mass, OCM), with representative episodes on October 22, 25, and 26.

**2. Section 2.1: some description of the vegetation type in the surrounding forest should be provided, so the reader can ascertain if other biogenic emissions such as monoterpenes were likely present that could influence the local chemistry.**

**Reply:**

Thanks for your suggestion. During the observation, the surrounding forest is lush around the YMK site. The vegetation type is evergreen broadleaf forests, which contributed to biogenic emissions. Previous literatures reported the monoterpene concentration in the YMK site, with a daily mean of 0.187 ppb (Zhu et al., 2021). Aboundant biogenic emissions will likely influence the local chemistry. We added the detailed description in Line 120-123.

**Revision:**

Line 120-123: Previous literatures reported the monoterpene concentration in the YMK site, with a daily mean of 0.187 ppb (Zhu et al., 2021). Aboundant biogenic emissions will likely influence the local chemistry.

**3. Line 132: '...via chemical transformation' add 'by addition of NO'. Please also state the purity of the NO and concentration of NO in the detection cell.**

**Reply:**

Thanks for your suggestion. The purity of the NO was mixed with 2% in N2, and the concentration of NO in the detection cell was  $\sim 1.6 \times 10^{12}$  cm-3. NO was passed through a ferrous sulfate filter to remove impurities (NO2, HONO, and so on) before being injected into the detection cell. We added the detailed description in Line 187-195.

**Revision:**

Line 187-195: For HO2 measurement, the NO gas was mixed with 2% in N2 to achieve HO2-to-OH conversion. NO was passed through a ferrous sulfate filter to remove impurities (NO2, HONO, and so on) before being injected into the detection cell. The NO concentration (~1.6 × 1012 cm-3) corresponding to a conversion efficiency of ~15% was selected to avoid RO2→HO2 interference (especially from RO2 radicals derived from long-chain alkanes (C  $\geq$  3), alkenes, and aromatic hydrocarbons). Previous study denoted that the percentage interference from alkene-derived RO2 under these operating conditions was no more than 5% (Wang et al., 2021).

**4. *Line 147: Is this a single pass laser configuration or multi-pass? Please state* **Reply:**

Thanks for your suggestion. The radical detection module utilized a single pass laser configuration, and the laser beam was amplified to a diameter of 8 mm. We stated the detailed description in Line 158-160.

**Revision:**

Line 158-160: The radical detection module utilized a single pass laser configuration, and the laser beam was amplified to a diameter of 8 mm.

**5. Line 155: change 'avoid' to 'reduce'**

**Reply:**

We followed the reviewer's comment. We changed the description in Line 167-170.

**Revision:**

Line 167-170: Efficient ambient air sampling was achieved using an aluminum nozzle (0.4 mm orifice), and the pressure in the chamber was maintained at 400 Pa via a vortex vacuum pump (XDS35i, Edwards) to reduce fluorescence quenching.

**6. Line 159: The ozone interference as a function of ambient ozone and $H_2O(v)$ concentration and laser power determined from the laboratory experiment should be provided.**

**Reply:**

Thanks for your suggestion. We added the detailed description in Line 173-177.

**Revision:**

Line 173-177: Due to the synchronous reaction at 308nm, wavelength modulation is not applicable to ozone photolysis interference. Through laboratory experiments, at 20 mW laser energy, every 1% water vapor concentration and 50 ppb ozone concentration can generate a  $2.5 \times 10^5$  cm-3 OH concentration. The results in this paper have subtracted the ozone photolysis interference (Fig. S2).

**Fig. S2.** Mean diurnal profiles of measured [OH] before (red line) and after (blue line) deducting the O3 interference. The coloured shadows denote the 25 and 75% percentiles. The grey areas denote nighttime.

7. Line 160 -164: The previous good agreement reported for OH measurements by this system and the PKU-LIF in a previous study doesn't translate to an interference-free OH observation in the present study. Chemical removal of ambient OH using an inlet pre-injector is now seen as standard practice for LIF OH instruments. In the absence of this, the authors should provide some information on the chemical environment (ozone, alkene, NOx concentrations) were the instrument was deployed during the intercomparison and explain how this contrasts with the current environmental conditions.

**Reply:**

Thanks for your suggestion. We will discuss whether internal interference exists in AIOFM-LIF from the following aspects:

First of all, literature research shows that measurement interference is more related to the length of the inlet in the low-pressure cell (Griffith et al., 2016). In terms of system design, the AIOFM-LIF system uses a short-length inlet design to minimize this and other unknown disturbances (The distance from radical sampling to fluorescence excitation is ~150 mm).

Additionally, potential interference may exist when the atmosphere contains abundant alkenes, ozone, and BVOCs, indicating that environmental conditions play leading roles in OH interferences (Mao et al., 2010; Fuchs et al., 2016; Novelli et al., 2014). An OH measurement comparison with a LIF instrument deployed an inlet preinjector (PKU-LIF), was conducted in a real atmosphere in a previous study (Zhang et al., 2022b). The ozonolysis interference on the measurement consistency of both systems was excluded under high-VOCs conditions. We have compared the chemical conditions during the intercomparison experiment and the current environmental conditions. Overall, the key parameters related to ozonolysis reactions ( $O_{3x}$  alkenes, isoprene and NOx) in YMK were similar to those during the comparison experiment, which is not conducive to generating potential OH interference. Therefore, it is not expected that OH measurement in the present study was affected by internal interference. We added the detailed description in Line 177-187.

| Species              | Intercomparison | YMK   |
|----------------------|-----------------|-------|
| O 3 (ppb) | 71.02           | 74.58 |
| Alkenes (ppb)        | 1.29            | 1.10  |
| Isoprene (ppb)       | 0.67            | 0.64  |
| NOx (ppb)            | 5.65            | 4.24  |

**Table.S1.** Comparison of key parameters related to ozonolysis reactions ( $O_3$ , alkenes, isoprene and NOx) between YMK and the intercomparison experiment. All the values are the diurnal average (10.00, 15.00)

**Revision:**

Line 177-187: In terms of system design, the AIOFM-LIF system incorporates a shortlength inlet design to minimize interferences from ozonolysis and other unknown factors (the distance from radical sampling to flourescence excitation is ~150 mm). An OH measurement comparison with an interference-free instrument, PKU-LIF, was conducted in a real atmosphere in a previous study (Zhang et al., 2022b). The ozonolysis interference on the measurement consistency of both systems was excluded under high-VOCs condition. Overall, the key parameters related to ozonolysis reactions (O3 alkenes isoprene and NOx) in YMK was similar to that during the intercomparison experiment, implies that the chemical conditions do not favor the generation of potential interference to OH measurement (Table S1).

**8. Line 166: The authors need to state the percentage interference from an alkenederived RO2 under these operating conditions (it won't be zero).**

**Reply:**

Thanks for your suggestion. In the previous work, we have calculated the conversion efficiency of alkene-derived RO2 to OH under different NO concentration (Wang et al., 2021). In the YMK observation, ethene accounted for about 70% of the total ethene concentration (Table S5). Therefore, we choose ethene and isoprene to investigate the percentage interference from an alkene-derived RO2. When NO was at  $1.6 \times 10^{12}$  cm-3, the conversion efficiency of HO2 was ~15%, and the percentage interference from ethene and isoprene-derived RO2 was 3.83% and 1.75%, respectively

(Wang et al., 2021). We added the detailed description in Line 193-195.

**Revision:**

Line 193-195: Previous study denoted that the percentage interference from alkenederived  $RO_2$  under these operating conditions was no more than 5% (Wang et al., 2021).

**9. Line 169: How much OH and HO2 was produced in the calibration? Was the calibration performed using a turbulent flow? How was the lamp flux determined? These details should be included in the manuscript.**

**Reply:**

Thanks for your suggestion. In the YMK observation, the calibration was performed in a laminar condition with a maximum flow rate of 20 SLM (standard liters per minute)(Wang et al., 2020; Wang et al., 2019). OH concentration was deduced by chemical radiometry according to the next Eq. :

$$[OH] = [HO_2] = \frac{1}{2} \cdot \frac{\sigma_{H_2O}}{\sigma_{O_2}} \cdot \frac{[H_2O]}{[O_2]} \cdot \frac{[O_3]}{P}$$

 $\sigma_{H_20}$  and  $\sigma_{O_2}$  represent the absorption cross-sections of water and oxygen, respectively. *P* represents the distribution factor of ozone concentration inside the tube. In previous studies, we obtained the actual measured values of  $\sigma_{O_2}$  and *P* through experiments (Wang et al., 2020). As the luminous flux in photolysis region is difficult to accurately measure, the linearly correlation between ozone concentration and 185 nm light flux was established. Ozone concentration in the flow tube was measured by a home-made Cavity Ring Down Spectrometer (CRDS, and the detection limit is 15 ppt@30 s, 1 $\sigma$ ). Mercury lamp intensity is adjusted to establish.

In the YMK campaign, the humidity varied between 40 – 80% (Fig. S3). In order to test different atmospheric conditions, both low (~40%) and high (~70%) levels of water vapor were selected to produce OH and HO2 radicals for calibration, and the corresponding HOx concentration obtained from the standard source was  $1.0 \times 10^9$  cm-3 and  $1.8 \times 10^9$  cm-3, respectively (Zhang et al., 2022b).

We added the detailed description in Line 196-212.

**Revision:**

Line 196-212: A standard HOx radical source was used to complete the calibration of the detection sensitivity (Wang et al., 2020). The radical source is based on the simultaneous photolysis of H2O/O2 by a 185 nm mercury lamp. Humidified air flow is introduced to produce equal amounts of OH and HO2 radicals after passing the photolysis region. The flow remained in a laminar condition with a maximum flow rate of 20 SLM (standard liters per minute). As the luminous flux in photolysis region is difficult to accurately measure, the linearly correlation between ozone concentration and 185 nm light flux was established. Ozone concentration in the flow tube was measured by a home-made Cavity Ring Down Spectrometer (CRDS, and the detection limit is 15 ppt(a)30 s, 1 $\sigma$ ). Mercury lamp intensity is adjusted to establish. The instrument was calibrated every 1 or 2 days (except for shutdown during rainy periods), and the sensitivity used for the data processing was an average of all of the calibration results. In the YMK campaign, the humidity varied between 40 - 80% (Fig. S3). In order to test different atmospheric conditions, both low (~40%) and high (~70%) levels of water vapor were selected to produce OH and HO2 radicals for calibration, and the corresponding HOx concentration obtained from the standard source was  $1.0 \times 10^9 \, \text{cm}^{-3}$ and  $1.8 \times 10^9$  cm-3, respectively (Zhang et al., 2022b).

**10. Section 2.2.2: Given the importance of HONO as an OH source in this study, some comments on possible instrument artefacts/or how interferences were corrected for should be discussed.**

**Reply:**

Thank you for your suggestion. HONO measurement was conducted using a commercial Long-Path Absorption Photometer (LOPAP). The LOPAP method utilizes two absorption tubes in series for differential correction, which effectively eliminates the influence of known interfering substances such as NO2 and N2O5, offering an advantage over traditional wet chemistry methods. This method has been extensively tested for its suitability in detecting HONO in complex atmospheric conditions, as demonstrated in previous studies by(Yang et al., 2022b; Yang et al., 2021b; Wang et al., 2023). During the YMK campaign, zero air measurements were taken every 8 hours for

a duration of 20 minutes to correct for instrument baseline fluctuations. Additionally, a liquid nitrite standard calibration was performed on a weekly basis to ensure the accuracy of the calibration curve used for measuring HONO concentrations. We added the detailed description in Line 225-233.

**Revision:**

Line 225-233: HONO measurement was conducted using a commercial Long-Path Absorption Photometer (LOPAP). The LOPAP method utilizes two absorption tubes in series for differential correction, which effectively eliminates the influence of known interfering substances such as NO2 and N2O5, offering an advantage over traditional wet chemistry methods. Zero air measurements were taken every 8 hours for a duration of 20 minutes to correct for instrument baseline fluctuations. This method has been extensively tested for its suitability in detecting HONO in complex atmospheric conditions, as demonstrated in previous studies by (Yang et al., 2022b; Yang et al., 2021b; Wang et al., 2023).

**11. Line 183: Which of the measured photolysis rates were used as model constraints?Reply:**

Thanks for your suggestion. Eight measured photolysis rates ( $j(NO_2)$ ,  $j(H_2O_2)$ , j(HCHO), j(HONO),  $j(NO_2)$ ,  $j(NO_3)$ ,  $j(O^1D)$ ) were used as model constraints.

**Revision:**

Line 233-234: Eight measured photolysis rates ( $j(NO_2)$ ,  $j(H_2O_2)$ , j(HCHO), j(HONO),  $j(NO_2)$ ,  $j(NO_3)$ ,  $j(O^1D)$ ) were used as model constraints.

**12. Line 204: Did the modelled OH and HO2 reach a steady state concentration during this time – what was the % difference between day 2 and day 3 radical concentrations?**

**Reply:**

We followed the reviewer's comment. We calculated the steady state concentrations of OH and HO2 radical using the Eq.(1)(2):

$$[OH]_{PSS} = \frac{j_{HONO}[HONO] + \varphi_{OH}j(O^{1}D)[O_{3}] + k_{HO_{2}+NO}[NO][HO_{2}]}{k_{OH}}$$
(1)

$$[HO_2]_{PSS} = \frac{k_{CO+OH}[CO][OH] + j_{HCHO}[HCHO] + k_{RO_2+NO}[NO][RO_2]}{k_{HO_2+NO}[NO]}$$
(2)

Due to the lack of RO2 radical observation data, substitute the  $[RO_2]$  and  $k_{OH}$  items in Eq.(1)(2) with the simulated value. The comparison of steady state and the modelled concentrations in the daytime were shown in Fig.3. During the entire observation period, the base model reached a steady state and showed good agreement with the calculated concentrations of OH and HO2 radicals using Eq.(1)(2). Specifically, on the second and third days, there were no significant differences between the steady-state calculation and the base model, with OH and HO2 concentration deviations of 7.5% and 3.1%, respectively.

Fig. 3. Timeseries of the observed and modelled parameters for OH, HO2 and  $k_{OH}$  during the observation period. (a) OH, (b) HO2, (c)  $k_{OH}$ .

**Revision:**

Line 262-267: In addition, another steady-state calculation method (PSS) can also be used to estimate the concentrations of OH and HO2 radicals (Eq. (1)(2), (Woodward-Massey et al., 2022; Slater et al., 2020)). Since the  $k_{OH}$  and RO2 concentrations were not obtained in this observation, simulated values are used as substitutes. Other radical and reactive intermediates are actual values that measured from the instruments in Table S2.

Line 369-374: Overall, the observed OH and HO2 concentrations were both well reproduced by the base model incorporating the RACM2-LIM1 mechanism. The

observed OH was underestimated only on the first days, and a slight model overestimation happened on October 23&24. PSS calculation showed good agreement with the base model, providing evidence of the balance of radical internal consistency in the daytime.

**13. Line 208: Could the authors explain their choice of the 18 hr lifetime? Were any model tests performed to assess how well the model predicted HCHO for example (if left unconstrained to HCHO) with an 18 hr lifetime? How sensitive were the modelled OH, HO2 and modelled kOH to the choice of this lifetime?**

**Reply:**

Thanks for your suggestion. After literature research, we found that the lifetime for the model-generated intermediate is usually set between 8 - 24 h (Ma et al., 2022; Yang et al., 2021a; Whalley et al., 2021). We tested the relationship between the first-order loss term and simulated OH, HO2,  $k_{OH}$  by changeing the lifetime (8h, 12h, 18h, and 24h). The results have been added to Fig. S4. The sensitivity analysis shows that when the lifetime changes within 8 - 24 hours (8h, 12h, 18h, and 24h). The values differed less than 5% between two cases for both OH, HO2,  $k_{OH}$ . Therefore, we finally chose a settling time of 18 hours, corresponding to first order loss rate of ~1.5 cm/s (by assuming a boundary layer height of about 1 km).

---

## Author Response (AR2)

Dear Editor and Referee,

Thanks for your suggestions which significantly help us to improve the manuscript. Hereby, we submit our responses and the manuscript has been revised accordingly. If there are any further questions or comments, please let us know.

Best regards

Renzhi Hu on behalf of all co-authors

Key Lab. of Environmental Optics & Technology, Anhui Institute of Optics and Fine Mechanics, Chinese Academy of Sciences

230031 Hefei China

E-mail: rzhu@aiofm.ac.cn

**Reviewer #1 (Minor Comments)**

1. *I just suggest changing 'NO gas was mixed with 2% in N2' to 'NO gas (2% in N2)'*

**Reply:**

Thanks for your suggestion. We modified the description in Line 187-188.

**Revision:**

Line 187-188: For $HO_2$ measurement, the NO gas (2% in $N_2$) was utilized to achieve $HO_2$-to-OH conversion.

2. *Change 'laser beam was amplified to a diameter of 8 mm' to 'laser beam had a diameter of 8 mm'*

**Reply:**

Thanks for your suggestion. We modified the description in Line 158-160.

**Revision:**

Line 158-160: The radical detection module utilized a single pass laser configuration, and the laser beam had a diameter of 8 mm.

3. *For the following question: '27. Fig. 10, line 532 - 539: Some further details on how the model was run when it was used to predict ozone are needed. What model constraints were changed to variables other than ozone (presumably NO2 was also changed to a variable)? ' If NO2 remained as a model constraint, this doesn't test the models ability to predict ozone as it is formed from the photolysis of NO2 (which was left constrained), so it isn't clear why the model predicted ozone changes when HONO is left unconstrained? Could additional model runs be performed where O3 and NO2 are unconstrained (constrained and unconstrained to HONO) as I think this would be a better test of the impact of HONO on O3?*

**Reply:**

Thanks for your suggestion. In the ozone-prediction mode, we use the 0-D

chemical box model incorporating a condensed mechanism, the regional atmospheric chemistry mechanism version 2-Leuven isoprene mechanism (RACM2-LIM1). The utilization of a box model significantly reduces the computational power and simplifies the transport processes to derive the chemical response efficiently. In the box model mechanism, no emission rates were input into the traditional box model, so the concentrations of NO, $NO_2$, and VOCs need to be constrained to measurements in order to explore the ozone-related chemical processes.

[Figure]

**Fig.** The $O_3$ concentration output by the ozone-prediction mode in different scenarios (Scenario 1: with input of $NO_2$, Scenario 2: without input of $NO_2$). The HONO concentration was constrained by measurement (w. Mea. HONO, red line) and unconstrained (w. O. HONO, blue line), respectively.

We attempted to follow the suggestions of the reviewers and did not constrain the concentrations of ozone and $NO_2$. The simulated results of ozone in this scenario are shown in the Figure above. The predicted daytime distribution of ozone concentration is basically the same in Scenario 1 (Fig.(a), with $NO_2$ constrained, adopted in this paper) and Scenario 2 (Fig.(b), without $NO_2$ constrained, recommended by the reviewer). However, due to the lack of sources in Scenario2, the box model cannot predict ozone concentrations normally without HONO input, and the obtained ozone concentration is approximately 0 (Fig.(b), blue line). The effect of HONO on ozone generation cannot be investigated under the condition that no $NO_2$ is input. Comparatively, removing the constraints on ozone and NO while keeping $NO_2$ as a constraint is a commonly used method in the box model for ozone prediction (Tan et al., 2018a). We added the description in the manuscript.

**Revision:**

Line 606-608&Supplement S1: Comparatively, removing the constraints on ozone and

NO while keeping $NO_2$ as a constraint is a commonly used method in the box model for ozone prediction (Tan et al., 2018a).

**Reviewer #2 (Minor Comments)**

1. *L44-46 : "Without the HONO constraint, simulated O3 decreased from ~75 ppb to a global background (~35 ppb), and daytime HONO concentrations were reduced to a low level (~70 ppt)." – Was HONO the only species that was unconstrained here? Or was O3 also unconstrained together with HONO? It is hard to believe that O3 decreased from 75 to 35 ppb, especially when the authors indicate on L41-42 that the ozone production rate increases by 33-39% when HONO is constrained.*

**Reply:**

Thanks for your suggestion. The ozone production rate increases by 33-39% when HONO is constrained. In the ozone prediction scenario, the constraints on observed ozone and NO concentrations were removed, upon the basis the effect of constrained/unconstrained HONO were investigated. We acknowledge the reviewer's opinion that the drop in ozone concentration from ~75 ppb to 35 ppb is somewhat strange, and suspect that there may be other factors related to nitrogen chemistry at play, making the change in ozone concentration without HONO input more complex. Therefore, and we emphasize that this is a result of sensitivity testing for ozone prediction. We conducted an additional test to investigate the changes in ozone concentration prediction and added a new Fig. S8 in the supplement. The HONO concentration was constrained by measurement (w. Mea. HONO, red line in Fig. S8), unconstrained (w. O. HONO, blue line in Fig. S8) and calculation (w. Cal. HONO, green line in Fig. S8). The calculated HONO concentration was limited to 2% of $NO_2$ concentration. This simple calculation method for HONO concentration has been validated in multiple field observations (Tan et al., 2019; Elshorbany et al., 2012). Compared to the condition without HONO input, the addition of calculated HONO concentration slightly improved the $O_3$ simulation effect, but the peak value remained at around 40 ppb. Therefore, the contribution of HONO to ozone concentration prediction has some rationality. We added the description in the manuscript.

[Figure]

**Fig. S8.** The O₃ concentrations simulated by the ozone-prediction mode. The HONO concentration was constrained by measurement (w. Mea. HONO, red line), unconstrained (w. O. HONO, blue line) and calculation (w. Cal. HONO, green line). The calculated HONO concentration was limited to 2% of NO₂ concentration.

**Revision:**

Line 608-611: Considering the complexity of HONO chemistry, this is more emphasized as a sensitivity test for ozone prediction, and its validity has been examined through simulated comparisons under different HONO concentrations (Fig. S8).

Line 44-46: In the ozone-prediction test, simulated O₃ decreased from ~75 ppb to a global background (~35 ppb) without the HONO constraint.

*2. L215-216: Is the measurement accuracy given as 1σ? Please indicate it in the text.*

**Reply:**

Thanks for your suggestion. We modified the description in Line 216-217.

**Revision:**

Line 216-217: At a typical laser power of 15 mW, the measurement accuracy for OH and HO₂ measurement was 13% and 17% (1σ), respectively.

*3. L527-528: "which also affects the quantum yield of NO2" – The reviewer does not understand what is meant here. Do the authors mean that it affects the amount of NO2 that is produced? Please clarify.*

**Reply:**

Thanks for your suggestion. Regarding the modification on Eq.8, it was suggested by Reviewer 2 (Question 18) during the discussion process. Considering the reaction

between $RO_2$ and NO, the formation of organic nitrates affects affects the amount of $NO_2$ that is produced. Therefore, we added $\alpha_i$ represents the organic nitrate yield in Eq.8 to determine the effective generation of $NO_2$ (Tan et al., 2018b). We modified the misleading description in Line 528-529.

**Revision:**

Line 528-529: $\alpha_i$ represents the organic nitrate yield, which affects the amount of $NO_2$ that is produced from the reaction between $RO_2$ and NO (Tan et al., 2018b).

*4. Eqs. 4: How did the authors derived this equation? The general equation to calculate the HO2 uptake rate is k=γ\*ASA\*γ(HO2)/4.*

**Reply:**

Thanks for your suggestion. The $HO_2$ uptake rate in Eq.4 was incorrectly written in the previous manuscript. We have modified the Equation in Line 283.

**Revision:**

Line 283:

$$k_{HO_2+uptake} = \frac{\gamma \times ASA \times v_{HO_2}}{4} \tag{4}$$

**Reviewer #2 (Edits)**

1. **L205: "Mercury lamp intensity is adjusted to establish." – Please clarify this sentence. It seems that something is missing.**

**Reply:**

Thanks for your suggestion. We have modified the sentence in Line 205-206.

**Revision:**

Line 205-206: Mercury lamp intensity is fine-tuned to establish a correlation between light intensity and ozone concentration.

2. **L208: "In the YMK c ampaign , the humidity varied between 40 80%" should read "In the YMK campaign , the relative humidity varied between 40 80%"**

**Reply:**

Thanks for your suggestion. We have modified the sentence in Line 209.

**Revision:**

Line 209: In the YMK campaign, the relative humidity varied between 40 – 80% (Fig. S3).

3. **L173-174: "Due to the synchronous reaction at 308nm , wavelength modulation is not applicable to ozone photolysis interference" should read "Since the ozone photolysis interference is due to the laser light itself, wavelength modulation does not allow removing it."**

**Reply:**

Thanks for your suggestion. We have modified the sentence in Line 173-174.

**Revision:**

Line 173-174: Since the ozone photolysis interference is due to the laser light itself, wavelength modulation does not allow removing it.

4. **L272: "diagnose the impacts of the reactive bromine chemistry." Should read "diagnose the impacts of the reactive bromine and iodine chemistry."**

**Reply:**

Thanks for your suggestion. We have modified the sentence in Line 271-273.

**Revision:**

Line 271-273: Considering the environmental characteristics of the MBL, the gas-phase mechanisms for bromine (Br) and iodine (I) were introduced into the base model to diagnose the impacts of the reactive bromine and iodine chemistry

5. *L527-528: "$\alpha i$ represents the side generation ratio of organic nitrate" – Should read "$\alpha i$ represents the organic nitrate yield".*

**Reply:**

Thanks for your suggestion. We have modified the sentence in Line 528-529.

**Revision:**

Line 528-529: $\alpha_i$ represents the organic nitrate yield, which affects the amount of $NO_2$ that is produced from the reaction between $RO_2$ and $NO$ (Tan et al., 2018b).

**References**

Elshorbany, Y. F., Steil, B., Brühl, C., and Lelieveld, J.: Impact of HONO on global atmospheric chemistry calculated with an empirical parameterization in the EMAC model, Atmos. Chem. Phys., 12, 9977-10000, 10.5194/acp-12-9977-2012, 2012.

Tan, Z., Lu, K., Jiang, M., Su, R., Wang, H., Lou, S., Fu, Q., Zhai, C., Tan, Q., Yue, D., Chen, D., Wang, Z., Xie, S., Zeng, L., and Zhang, Y.: Daytime atmospheric oxidation capacity in four Chinese megacities during the photochemically polluted season: a case study based on box model simulation, Atmos Chem Phys, 19, 3493-3513, 10.5194/acp-19-3493-2019, 2019.

Tan, Z. F., Lu, K. D., Jiang, M. Q., Su, R., Dong, H. B., Zeng, L. M., Xie, S. D., Tan, Q. W., and Zhang, Y. H.: Exploring ozone pollution in Chengdu, southwestern China: A case study from radical chemistry to O3-VOC-NOx sensitivity, Sci Total Environ, 636, 775-786, 10.1016/j.scitotenv.2018.04.286, 2018a.

Tan, Z. F., Lu, K. D., Dong, H. B., Hu, M., Li, X., Liu, Y. H., Lu, S. H., Shao, M., Su, R., Wang, H. C., Wu, Y. S., Wahner, A., and Zhang, Y. H.: Explicit diagnosis of the local ozone production rate and the ozone-NOx-VOC sensitivities, Sci. Bull., 63, 1067-1076, 10.1016/j.scib.2018.07.001, 2018b.

---

## Author Response (AR3)

Dear Editor,

    Thanks for your suggestions which significantly help us to improve the manuscript. Hereby, we submit our responses and the manuscript has been revised accordingly. If there are any further questions or comments, please let us know.

Best regards

Renzhi Hu on behalf of all co-authors

Key Lab. of Environmental Optics & Technology, Anhui Institute of Optics and Fine Mechanics, Chinese Academy of Sciences

230031 Hefei China

E-mail: rzhu@aiofm.ac.cn

**Editor (Minor Comments)**

*1. About the ozone predicting model calculations:*

*Considering the modelled O3 levels start decreasing from about 1500LT, it is very likely that some loss term is included, while the authors stated that the impacts of vertical entrainment and horizontal advection were (in general) ignored (lines 611-612). On the other hand, the authors also state in lines 593-594 that O3 was modelled with a first order loss term. Clarification is necessary. Also, the deposition process with a relatively long lifetime (15 hours) will not be sufficient to start to decrease O3 levels as early as 1500LT (as photochemical production goes on). Careful description is needed.*

**Reply:**

Thanks for your suggestion. First, we verified the data and found that the simulated peak ozone concentration occurred around 16:00 and started to decrease afterwards. We acknowledge the editor's point that there are some ozone loss processes existed, such as vertical entrainment and horizontal advection. The loss of model-generated $O_3$ by deposition or mixing was represented as a first-order deposition rate corresponding to a lifetime of 18 hours. We clarified the description in Lines 588&604-606.

[Figure]

Furthermore, we also investigated the influence of different deposition rates on the simulated ozone concentration. We constrained the lifetimes to 8 hours, 18 hours, and 36 hours, which correspond to first-order loss rates of approximately 3.5 cm/s, 1.5 cm/s, and 0.8 cm/s, respectively, assuming a boundary layer height of about 1 km. Under different lifetimes, the changes in the rate of ozone concentration ($d(O_3)/dt$) did not

exceed 3 ppb/h, as the Figure above. Therefore, the influence of different deposition rates in the short term can be negligible compared to the chemical processes of ozone production.

[Figure]

It is crucial to consider the variability in the distribution of ozone concentration, as there are changes in deposition processes, as pointed out by the editor. With a deposition time of 18 hours, the peak ozone concentration shifted to around 16:00. As shown in the Figure above, from the perspective of photochemistry, the rate of ozone generation significantly decreases to 1-2 ppb/h after 16:00. The combined effects of other loss processes and attenuated photochemical processes contribute to a certain decrease in ozone concentration.

**Revision:**

Line 588: The deposition or mixing processes were equivalent to a lifetime of 18 hours to all species.

Line 604-606: The loss of model-generated $O_3$ by deposition or mixing was represented as a first-order deposition rate corresponding to a lifetime of 18 hours.

---

## Author Response (AR4)

Dear Editor,

Thanks for your suggestions which significantly help us to improve the manuscript. Hereby, we submit our responses and the manuscript has been revised accordingly. If there are any further questions or comments, please let us know.

Best regards

Renzhi Hu on behalf of all co-authors

Key Lab. of Environmental Optics & Technology, Anhui Institute of Optics and Fine Mechanics, Chinese Academy of Sciences

230031 Hefei China

E-mail: rzhu@aiofm.ac.cn

**Editor (Minor Comments)**

1. *Your first figure inserted to the Reply file shows that the d[O3]/dt becomes NEGATIVE from around 1700LT (or as early as 1500LT). Which process is driving this? Why bringing down to -5 ppb h-1 throughout midnight? NO titration or ground-surface dry deposition? If you take into account diurnally-varying boundary layer height (i.e. shallow layer in the night), please describe this.*

   *Then your second figure here shows "P(O3)", kept positive throughout the night. Is it GROSS or NET production from photochemistry? Just by adding dry deposition, do you have the quantity you showed as d[O3]/dt in the first figure?*

**Reply:**

Thanks for your suggestion. In this figure, we did not consider the diurnally-varying boundary layer height into the box model. Regarding the decrease in $d(O_3)/dt$ around 17:00, we believe it is the result of the combined effects of attenuated photochemistry and dry deposition after 17:00. Similar diurnal profiles have been observed in other sites (Tan et al., 2019). To further investigate the influence of NO titration, we conducted tests based on the change rates of Ox concentration, i.e., $d(Ox)/dt$ (Figure 1(a) below, the difference from the previous version is that we have corrected the time offset for $d(Ox)/dt$). The results showed that $d(Ox)/dt$ did not exhibit significant differences compared to $d(O_3)/dt$. Therefore, NO titration is not the main reason for the $O_3$ concentration change.

[Figure]

**Figure 1(a)** The change rates of Ox concentration (d(Ox)/dt) at different dilution times **(b)** The removal pathways of Ox over an 18-hour dilution time.

Regarding decrease trend of d(Ox)/dt of -5 ppb/h around midnight, we have

conducted additional discussions on the removal pathways of Ox over an 18-hour dilution time (the Figure 1(b) above). The "Deposition" term includes the removal of $NO_2$ and ozone through deposition, the "Ozonolysis" term involves the $O_3$+alkenes reactions, and the "$j(O_3)$ " term represents the photolysis channel of ozone. The "Others" term represents the removal processes of Ox through reactions with other species (such as the reactions between $O_3$ and OH, $HO_2$). The results indicate that the deposition process is the main pathway for Ox removal during the nighttime.

[Figure]

**Figure 2 (a)** The diurnal profile of modelled ozone concentration and P(Ox). **(b)** The deposition pathway of Ox.

Regarding your second question, the "ozone production" mentioned in the second figure should refer to the net production of total oxidants, which is represented as P(Ox). We updated the image label (Figure 2(a)) to reflect this correction. We have now included the calculation of P(Ox) based on the ozone simulation scenario, specifically considering the input of $NO_2$ without inputting NO and $O_3$. This calculation is represented by the green bars in Figure 2(a). After adding the dry deposition in Figure 2(b), the quantity of ozone change is similar with the d(Ox)/dt in Figure 1(a).

**Reference**

Tan, Z., Lu, K., Jiang, M., Su, R., Wang, H., Lou, S., Fu, Q., Zhai, C., Tan, Q., Yue, D., Chen, D., Wng, Z., Xie, S., Zeng, L., and Zhang, Y.: Daytime atmospheric oxidation capacity in four Chinese megacities during the photochemically polluted season: a case study based on box model simulation, Atmos Chem Phys, 19, 3493-3513, 10.5194/acp-19-3493-2019, 2019.